# MIRROR MEAN-FIELD LANGEVIN DYNAMICS

## ABSTRACT

The mean-field Langevin dynamics (MFLD) minimizes an entropy-regularized nonlinear convex functional on the Wasserstein space over $\mathbb{R}^d$, and has gained attention recently as a model for the gradient descent dynamics of interacting particle systems such as infinite-width two-layer neural networks. However, many problems of interest have constrained domains, which are not solved by existing mean-field algorithms due to the global diffusion term. We study the optimization of probability measures constrained to a convex subset of $\mathbb{R}^d$ by proposing the *mirror mean-field Langevin dynamics* (MMFLD), an extension of MFLD to the mirror Langevin framework. We obtain linear convergence guarantees for the continuous MMFLD via a uniform log-Sobolev inequality, and uniform-in-time propagation of chaos results for its time- and particle-discretized counterpart.

## 1 INTRODUCTION

In this work, we study the problem of minimizing an entropy-regularized functional on the space of probability measures constrained to a convex subset $\mathcal{X} \subseteq \mathbb{R}^d$:

$$\mathcal{L}(\mu) := F(\mu) + \lambda \mathsf{Ent}(\mu), \tag{1}$$

where $\mu \in \mathcal{P}_2(\mathcal{X})$, $F : \mathcal{P}_2(\mathcal{X}) \to \mathbb{R}$ is a linearly convex functional, and $\mathsf{Ent}(\mu) = \int \log \frac{d\mu}{dx} d\mu$ is the entropy of $\mu$. Such problems arise naturally in statistical inference and machine learning, such as when studying infinite-width limits of neural networks (Mei et al., 2018; Nitanda et al., 2022; Suzuki et al., 2023c; Takakura & Suzuki, 2024), tensor decomposition (Chizat & Bach, 2018), sparse spikes deconvolution (Chizat & Bach, 2018), density estimation (Suzuki et al., 2023b) and discrepancy minimization (Gretton et al., 2012; Suzuki et al., 2023b). Often, a nonconvex optimization problem of a particle system can be formulated as an instance of (1) by lifting to the space of measures.

In the simplest case, we wish to sample from a distribution $\mu_* \propto e^{-f/\lambda}$ for a potential $f : \mathcal{X} \to \mathbb{R}$, which corresponds to (1) where $F(\mu) = \int f d\mu$ is linear. When $\mathcal{X} = \mathbb{R}^d$, a classical approach is to employ a discretization of the Langevin dynamics

$$dX_t = -\nabla f(X_t)dt + \sqrt{2\lambda}dB_t, \tag{2}$$

where $B_t$ is standard Brownian motion. The trajectory $\mu_t = \mathrm{Law}(X_t)$ can be interpreted as the Wasserstein gradient flow minimizing the KL divergence with respect to $\mu_*$ (Jordan et al., 1998); this remarkable connection has led to many fruitful developments in both sampling and optimization, see for instance Wibisono (2018); Chewi (2023) for an exposition. The convergence of (2) in various metrics (Wasserstein, TV, KL, $\chi^2$, Rényi) has been extensively analyzed under strong log-concavity or functional inequalities such as log-Sobolev or Poincaré inequalities (Wibisono, 2018; 2019; Vempala & Wibisono, 2019; Durmus & Moulines, 2017; 2019; Li et al., 2021; Chewi et al., 2022; Mousavi-Hosseini et al., 2023).

In the constrained or ill-conditioned case, however, (2) cannot be applied as the diffusion term will cause mass to escape $\mathcal{X}$. This problem can be solved by imposing an alternative non-flat geometry on $\mathcal{X}$ and running the mirror Langevin dynamics (MLD), the analogue of mirror descent for sampling (Hsieh et al., 2018; Zhang et al., 2020). By introducing a barrier function $\phi : \mathcal{X} \to \mathbb{R}$, the mirror map $\nabla \phi$ induces an isometry between the primal space $\mathcal{X}$ with the Hessian metric $\nabla^2 \phi$, and the dual space $\mathbb{R}^d$ with the metric $\nabla^2 \phi^*$. This allows us to apply a transformed diffusion in the dual space:

$$X_t = \nabla \phi^*(Y_t), \quad dY_t = -\nabla f(X_t)dt + \sqrt{2\lambda \nabla^2 \phi(X_t)}dB_t. \tag{3}$$

The convergence of (3) under a mirror Poincaré inequality have been established in Chewi et al. (2020) and various discretization schemes have been analyzed in Zhang et al. (2020); Jiang (2021); Ahn & Chewi (2021); Li et al. (2022). In particular, Ahn & Chewi (2021) proposed a discretized mirror Langevin algorithm which ensures vanishing bias under mild assumptions on the mirror map.

Returning to the general problem (1), when $\mathcal{X} = \mathbb{R}^d$, this can be solved efficiently by running the mean-field Langevin dynamics (MFLD) (Nitanda et al., 2022; Chizat, 2022) extending (2), given by the McKean-Vlasov process

$$dX_t = -\nabla \frac{\delta F(\mu_t)}{\delta \mu}(X_t)dt + \sqrt{2\lambda}dB_t, \tag{4}$$

which is the Wasserstein gradient flow minimizing $\mathcal{L}$. The convergence of (4) and its time discretization has been established under a log-Sobolev inequality (LSI) in (Nitanda et al., 2022). Following works have studied *propagation of chaos* (Sznitman, 1991), that is, the approximation error induced when $\mu_t$ is replaced by a finite particle system. A dedicated line of work has obtained error bounds which converge to zero uniformly for all time as the number of particles $N \to \infty$ by leveraging uniform LSI (Chen et al., 2023; Suzuki et al., 2023a;b). Recent works have further shown that these errors can be made independent of the LSI constant (Nitanda, 2024; Chewi et al., 2024; Nitanda et al., 2025).

The same analyses may be applied if $\mathcal{X}$ is a complete Riemannian manifold without boundary, e.g., a hypersphere or torus. However, the problem for general $\mathcal{X}$ has remained open. Nonetheless, many applications of MFLD require the domain of optimization to be (implicitly or explicitly) constrained, e.g., trajectory inference (Chizat et al., 2022; Gu et al., 2025a;b), computation of Wasserstein barycenters (Chizat, 2023; Vaskevicius & Chizat, 2023), computation of discrepancy measures (Suzuki et al., 2023b) or signal deconvolution (Chizat & Bach, 2018) with bounded support, and optimization of neural networks (Nitanda et al., 2022; 2025) with constrained parameters. A more detailed overview of related works is provided in Appendix A.

### 1.1 CONTRIBUTIONS

We propose the *mirror mean-field Langevin dynamics* (MMFLD), which unifies the mirror (3) and mean-field (4) analyses to solve the constrained distributional optimization problem (1) for general convex $\mathcal{X} \subseteq \mathbb{R}^d$ and (linear) convex $F : \mathcal{P}_2(\mathcal{X}) \to \mathbb{R}^d$. To the best of our knowledge, this constitutes the first algorithm minimizing (1) with global convergence guarantees. In particular, we provide non-asymptotic convergence rates for both the continuous-time mean-field flow (Section 3), and the time- and particle-discretized algorithm (Section 4). Crucially, we show the recent advances in propagation of chaos analysis extend gracefully to constrained domains.

**Notation.** The Euclidean norm on $\mathbb{R}^d$ is denoted as $\| \cdot \|$. We use $\int f$ to denote the integral with respect to Lebesgue measure: $\int f \, dx$. When the integral is with respect to a different measure $\mu$, we write it as $\int f \, d\mu$. We use $X, Y$ for random variables and non-tilde $\mu$ and tilde $\tilde{\mu}$ for probability distributions in the primal and dual spaces, respectively. When it is clear from context, we will sometimes abuse notation by identifying a measure with its density. We also use the same symbol $C_{\mathsf{LSI}}$ to denote log-Sobolev constants for LSI, mirror LSI, and uniform-in-$N$ mirror LSI, etc. $\delta_x$ denotes the Dirac delta measure at $x \in \mathbb{R}^d$. For a measurable function $T : \mathcal{X} \to \mathcal{Y}$, we use $T_\sharp \mu$ to denote the pushforward of $\mu$ by $T$, e.g. $T_\sharp \mu(B) = \mu(T^{-1}(B))$ for any measurable set $B \subseteq \mathcal{Y}$. Finally, we use $\mathcal{P}_2(\mathcal{X})$ to denote the set of Borel probability measures over $\mathcal{X}$ with finite second moments, equipped with the 2-Wasserstein metric.

## 2 PRELIMINARIES

In order to develop our main object of interest, the mirror mean-field Langevin dynamics, we first provide discussion of the mirror Langevin dynamics and mean-field Langevin dyamics as a primer.

### 2.1 MIRROR LANGEVIN DYNAMICS

The mirror Langevin dynamics (MLD) has been extensively studied in the context of constrained sampling problems (Zhang et al., 2020; Jiang, 2021; Li et al., 2022). Suppose we want to sample

from a probability distribution $\mu$ supported on a convex set $\mathcal{X} \subseteq \mathbb{R}^d$. We assume $\mu$ is absolutely continuous with respect to Lebesgue measure on $\mathbb{R}^d$ and has density $\mu \propto e^{-f/\lambda}$ for some differentiable $f : \mathcal{X} \to \mathbb{R}$.

Certainly, one way is to run the Langevin dynamics (or algorithm) (2) and project onto $\mathcal{X}$. However, one undesirable behavior of this projection step is that this will necessarily put positive mass onto $\partial \mathcal{X}$; see the numerical experiments in Section 5. Alternatively, we can enforce the distribution to stay in $\mathcal{X}$ by changing the geometry of the underlying space.

Towards that end, let $\phi : \mathcal{X} \to \mathbb{R}$ be a thrice-differentiable strictly convex function of Legendre type (Rockafellar, 1997). We require $\|\nabla \phi(x)\| \to \infty$ and $\nabla^2 \phi(x) \to \infty$ as $x$ approaches $\partial \mathcal{X}$, as this ensures the diffusion remains inside the domain $\mathcal{X}$. We call $\nabla \phi : \mathcal{X} \to \mathbb{R}^d$ the *mirror map* and $\mathcal{Y} = \nabla \phi(\mathcal{X}) = \mathbb{R}^d$ the *dual space*. Further, define $\phi^* : \mathbb{R}^d \to \mathbb{R}$ to be the Legendre dual (convex conjugate) of $\phi$, given as $\phi^*(y) = \sup_{x \in \mathcal{X}} \langle x, y \rangle - \phi(x)$. In Appendix C, we provide discussion on properties of $\phi$.

The mirror Langevin dynamics (MLD) in primal space satisfies the SDE

$$\begin{cases} X_t & = \nabla \phi^*(Y_t) \\ dY_t & = -\nabla f(X_t) dt + \sqrt{2\lambda \nabla^2 \phi(X_t)} dB_t, \end{cases} \tag{5}$$

and its density $\mu_t = \text{Law}(X_t)$ satisfies the Fokker-Planck PDE

$$\frac{\partial \mu_t}{\partial t} = \lambda \nabla \cdot \left( \mu_t [\nabla^2 \phi]^{-1} \nabla \log \frac{\mu_t}{\mu} \right).$$

This is equivalent to the Riemmanian Langevin dynamics in the primal space with metric given by the Hessian $\nabla^2 \phi$, or in the dual space with metric $\nabla^2 \phi^*$, as $(\mathcal{X}, \nabla^2 \phi)$ is isometric to $(\mathbb{R}^d, \nabla^2 \phi^*)$. This ensures we obtain the same the convergence guarantees in both the primal and dual spaces.

We will require standard Lipschitz conditions on $f$ (Øksendal, 2010) and $\|[\nabla^2 \phi(x)]^{1/2} - [\nabla^2 \phi(x')]^{1/2}\|_{\mathsf{F}} \leq O(\|x - x'\|_2)$ for every $x, x' \in \mathcal{X}$. This ensures well-posedness of (5); see Appendix A of Zhang et al. (2020) for more details.[1]

Using Itô's formula, we can obtain a representation of (5) purely in terms of $X_t$; however, it depends on the third-derivative of $\phi$, see (Chewi, 2023, Exercise 10.1). Alternatively, we can reformulate the MLD entirely in the dual space as the following SDE:

$$dY_t = -\nabla f(\nabla \phi^*(Y_t)) dt + \sqrt{2\lambda [\nabla^2 \phi^*(Y_t)]^{-1}} dB_t.$$

The distribution $\tilde{\mu}_t = \text{Law}(Y_t)$ is the pushforward of $\mu_t$ under the mirror map: $\tilde{\mu}_t = (\nabla \phi)_{\sharp} \mu_t$.

To initiate the isoperimetric analysis, we first introduce the entropy, Kullback-Leibler (KL) divergence, and relative Fisher information. Denote the entropy of a nonnegative functional $f \geq 0$ as $\mathsf{Ent}_\mu(f) := \mathbb{E}_\mu[f \log f] - \mathbb{E}_\mu[f^2] \log \mathbb{E}_\mu[f^2]$, KL divergence between two measures $\mu, \nu$ as $\mathsf{KL}(\mu \| \nu) := \mathbb{E}_\nu \left[ \frac{\mu}{\nu} \log \frac{\mu}{\nu} \right] = \int \mu \log \frac{\mu}{\nu}$, and the relative Fisher information (modified by the mirror map) as $\mathsf{FI}_\phi(\mu \| \nu) := \mathbb{E}_\mu \left\langle \nabla \log \frac{\mu}{\nu}, [\nabla^2(\phi)]^{-1} \nabla \log \frac{\mu}{\nu} \right\rangle$.[2]

We require the stationary distribution $\mu_*$ to satisfy a mirror log-Sobolev inequality:

**Assumption 1** (Mirror log-Sobolev inequality). *There exists a constant $C_{\mathsf{LSI}} > 0$ such that $\mu_*$ satisfies a mirror log-Sobolev inequality with constant $C_{\mathsf{LSI}}$, that is, for any smooth function $g : \mathbb{R}^d \to \mathbb{R}$, we have $\mathsf{Ent}_{\mu_*}(g^2) \leq \frac{2}{C_{\mathsf{LSI}}} \mathbb{E}_{\mu_*} \left[ \|\nabla g(x)\|^2_{[\nabla^2 \phi(x)]^{-1}} \right]$. Equivalently, by setting $g := \sqrt{\frac{d\mu}{d\mu_*}}$, for every $\mu \in \mathcal{P}_2(\mathcal{X})$, it holds that $\mathsf{KL}(\mu \| \mu_*) \leq \frac{1}{2C_{\mathsf{LSI}}} \mathsf{FI}_\phi(\mu \| \mu_*)$.*

A simple method to verify the mirror LSI is as follows: the mirror LSI is satisfied with constant $C_0/\alpha$ if the classical LSI is satisfied with constant $C_0$ and $\phi$ is $\alpha$-strongly convex (Daaloul et al., 2025). We provide the proof, an explicit family of distributions satisfying the mirror LSI, and some properties of the mirror LSI in Appendix B. Under this assumption, we can prove the following convergence guarantee.

---

[1] In the sequel, we will also implicitly assume this Frobenius norm bound for all of our results on the mirror mean-field Langevin dynamics as well.

[2] In the Euclidean setting, we drop the dependence on the mirror map and use $\mathsf{FI}(\mu \| \nu)$.

**Theorem 2.1.** *Let $\{\mu_t\}_{t\geq 0}$ denote the evolution of (5) and $\mu_*$ denote its stationary distribution. Under Assumption 1, for $t \geq 0$, it holds that*

$$\mathsf{KL}(\mu_t \,\|\, \mu_*) \leq e^{-2C_{\mathsf{LSI}}\lambda t}\mathsf{KL}(\mu_0 \,\|\, \mu_*).$$

A proof is given in Jiang (2021), but for completeness, we provide a short proof in Appendix C.2.

## 2.2 Mean-Field Langevin Dynamics

Suppose we want to minimize the following entropy-regularized functional:

$$\mathcal{L}(\mu) := F(\mu) + \lambda\mathsf{Ent}(\mu), \tag{6}$$

where $F : \mathcal{P}_2(\mathbb{R}^d) \to \mathbb{R}$ is a linearly convex functional (Definition 2), $\mathsf{Ent}(\mu) := \int \log \frac{d\mu}{dx} d\mu$ is the entropy of $\mu$, and $\lambda > 0$ is the temperature parameter. We start by defining the first variation of $F$.

**Definition 1** (First variation). *We say $F : \mathcal{P}_2(\mathbb{R}^d) \to \mathbb{R}$ admits a first variation at $\mu \in \mathcal{P}_2(\mathbb{R}^d)$ if there exists a continuous function $\frac{\delta F(\mu)}{\delta\mu} : \mathbb{R}^d \to \mathbb{R}$ such that for any $\mu, \mu' \in \mathcal{P}_2(\mathbb{R}^d)$, we have*

$$\left.\frac{dF(\mu + \epsilon(\mu' - \mu))}{d\epsilon}\right|_{\epsilon=0} = \int \frac{\delta F(\mu)}{\delta\mu}(\mu' - \mu)dx. \tag{7}$$

*If the first variation of $F$ exists, it is unique up to a constant.*

To solve (6), we consider the mean-field Langevin dynamics (MFLD):

$$dX_t = -\nabla\frac{\delta F(\mu_t)}{\delta\mu}(X_t)dt + \sqrt{2\lambda}dB_t, \tag{8}$$

a McKean-Vlasov process where $\mu_t = \mathrm{Law}(X_t)$ and $B_t$ is the standard Brownian motion in $\mathbb{R}^d$. It follows that $\mu_t$ solves the nonlinear Fokker-Planck PDE

$$\frac{\partial\mu_t}{\partial t} = \nabla\cdot\left(\mu_t\nabla\frac{\delta F(\mu_t)}{\delta\mu}\right) + \lambda\Delta\mu_t = \lambda\nabla\cdot\left(\mu_t\nabla\log\frac{\mu_t}{\hat{\mu}_t}\right), \tag{9}$$

where $\hat{\mu}_t$ is the proximal Gibbs distribution associated to $\mu_t$ (see Definition 3 below). To ensure existence and convergence, the following assumptions are required.

**Assumption 2** (Lipschitz and smoothness). *There exist constants $M_1, M_2 > 0$ such that for any $\mu, \mu' \in \mathcal{P}_2(\mathbb{R}^d)$, $x, x' \in \mathbb{R}^d$, it holds that $\left\|\nabla\frac{\delta F(\mu)}{\delta\mu}(x)\right\|_2 \leq M_1$ and*

$$\left\|\nabla\frac{\delta F(\mu)}{\delta\mu}(x) - \nabla\frac{\delta F(\mu')}{\delta\mu}(x')\right\|_2 \leq M_2(W_2(\mu, \mu') + \|x - x'\|_2).$$

We briefly remark that the first part of the assumption is only required for the discretization analysis. Here $W_2(\mu, \mu')$ is the 2-Wasserstein distance between measures $\mu$ and $\mu'$ in $\mathcal{P}_2(\mathcal{X})$,

$$W_2^2(\mu, \mu') := \inf_{\pi\in\Pi(\mu,\mu')}\int_{\mathcal{X}\times\mathcal{X}}\|x - x'\|^2 d\pi(x, x'),$$

where $\Pi(\mu, \mu') := \{\pi \in \mathcal{P}(\mathcal{X} \times \mathcal{X}) \mid (P_x)_\sharp\pi = \mu, (P_y)_\sharp\pi = \mu'\}$ is the set of transport plans, and $P_x(x, y) := x, P_y(x, y) := y$ are the projections onto the first and second coordinates, respectively.

**Definition 2.** *We say a functional $F : \mathcal{P}_2(\mathbb{R}^d) \to \mathbb{R}$ is linearly convex if for every $\mu, \nu \in \mathcal{P}_2(\mathbb{R}^d)$ and $\alpha \in [0, 1]$, it holds that $F(\alpha\mu + (1 - \alpha)\nu) \leq \alpha F(\mu) + (1 - \alpha)F(\nu)$.*

**Assumption 3.** *$F$ is linearly convex and (6) admits a minimizer $\mu_*$.*

For example, this holds if $F$ is of the form $\sum_i \ell_i\left(\int f_i d\mu\right)$ for convex losses $\ell_i$. Then the following can be shown:

**Theorem 2.2.** *Under Assumptions 2 and 3, the minimizer $\mu_*$ of (6) is unique and its density satisfies $\mu_* \propto \exp\left(-\frac{1}{\lambda}\frac{\delta F(\mu_*)}{\delta\mu}\right)$.*

The convergence analysis of the MFLD relies on the proximal Gibbs distribution (Nitanda et al., 2022; Chizat, 2022).

**Definition 3** (Proximal Gibbs distribution). *For each $\mu \in \mathcal{P}_2(\mathbb{R}^d)$, we define $\hat{\mu}$ to be the Gibbs distribution such that $\hat{\mu} \propto \exp\left(-\frac{1}{\lambda}\frac{\delta F(\mu)}{\delta \mu}\right)$.*

Note that for the *linear* Langevin dynamics, the proximal Gibbs distribution coincides with the stationary distribution $\mu_*$. Next, we introduce the *uniform* log-Sobolev inequality.

**Assumption 4** (Uniform LSI). *Suppose there exists a constant $C_{\mathsf{LSI}} > 0$ such that for every $\mu \in \mathcal{P}_2(\mathbb{R}^d)$, the proximal Gibbs distribution $\hat{\mu}$ satisfies a log-Sobolev inequality with constant $C_{\mathsf{LSI}}$, that is, $\mathsf{KL}(\mu \,\|\, \hat{\mu}) \leq \frac{1}{2C_{\mathsf{LSI}}}\mathsf{FI}(\mu \,\|\, \hat{\mu})$.*

In the mean-field setting, this assumption is generally verified via the Holley-Stroock perturbation technique (Holley & Stroock, 1987) from the regularization term $\lambda$; see Suzuki et al. (2023b) for more discussion. Nitanda et al. (2022); Chizat (2022) use this key ingredient, along with the entropy sandwich inequality (Lemma C.2), to prove the convergence of the MFLD.

**Theorem 2.3.** *Let $(\mu_t)_{t\geq 0}$ be the evolution described by (9). Under Assumptions 2-4, it holds that*
$$\mathcal{L}(\mu_t) - \mathcal{L}(\mu_*) \leq e^{-2C_{\mathsf{LSI}}\lambda t}(\mathcal{L}(\mu_0) - \mathcal{L}(\mu_*)).$$

The discretization of MFLD has also been studied in Nitanda et al. (2022); Suzuki et al. (2023b); Nitanda (2024); we refer to these works for a detailed analysis.

## 3 MIRROR MEAN-FIELD LANGEVIN DYNAMICS

Now with the necessary technical background, we introduce our main object of study. Suppose we want to minimize (6) subject to the additional constraint that $\mu \in \mathcal{P}_2(\mathcal{X})$ for a convex set $\mathcal{X} \subseteq \mathbb{R}^d$:
$$\underset{\mu\in\mathcal{P}_2(\mathcal{X})}{\arg\min}\, \mathcal{L}(\mu) := F(\mu) + \lambda\mathsf{Ent}(\mu), \tag{10}$$
where the domain of $F$ and its first-variation are accordingly modified from (6), e.g., now we have $F : \mathcal{P}_2(\mathcal{X}) \to \mathbb{R}$, and its first variation at $\mu \in \mathcal{P}_2(\mathcal{X})$ is a function $\frac{\delta F(\mu)}{\delta \mu} : \mathcal{X} \to \mathbb{R}$. As we remarked in the introduction, restriction to a convex domain is a natural and frequently studied constraint in various optimization problems (Chizat et al., 2022; Chizat, 2023; Vaskevicius & Chizat, 2023; Gu et al., 2025a).

For this problem, we introduce the *mirror mean-field Langevin dynamics* (MMFLD), which is defined as the following (primal) SDE over $\mathcal{X}$:
$$\begin{cases} X_t &= \nabla\phi^*(Y_t) \\ dY_t &= -\nabla\frac{\delta F(\mu_t)}{\delta\mu}(X_t)dt + \sqrt{2\lambda\nabla^2\phi(X_t)}dB_t, \end{cases} \tag{11}$$
where $\mu_t = \mathrm{Law}(X_t)$. Similar to the mirror Langevin and mean-field Langevin settings, it is easy to check that (12) corresponds to the following Fokker-Planck PDE:
$$\frac{\partial}{\partial t}\mu_t = \lambda\nabla\cdot\left(\mu_t[\nabla^2\phi]^{-1}\nabla\log\frac{\mu_t}{\hat{\mu}_t}\right). \tag{12}$$

Next, we introduce the following notions of local and dual norms, which allows us to define the notion of relative Lipschitz and smoothness.

**Definition 4** (Local and dual norms). *Given a $C^2$ strictly convex function $\phi$, the local norm at $x \in \mathrm{int}\,\mathrm{dom}\,\phi$ with respect to $\phi$ is defined as $\|u\|_{\nabla^2\phi(x)} = \langle u, \nabla^2\phi(x)u\rangle^{1/2}$. The local dual norm is similarly defined as $\|u\|_{[\nabla^2\phi(x)]^{-1}} = \langle u, [\nabla^2\phi(x)]^{-1}u\rangle^{1/2}$.*

**Assumption 5** (Relative Lipschitz and smoothness). *There exists constants $M_1, M_2 > 0$ such that for any $\mu, \mu' \in \mathcal{P}_2(\mathcal{X})$, $x, x' \in \mathcal{X}$, $\frac{\delta F(\mu)}{\delta\mu}$ is differentiable with $\left\|\nabla\frac{\delta F(\mu)}{\delta\mu}(x)\right\|_{[\nabla^2\phi(x)]^{-1}} \leq M_1$ and*

$$\left\|\nabla\frac{\delta F(\mu)}{\delta\mu}(x) - \nabla\frac{\delta F(\mu')}{\delta\mu}(x')\right\|_{[\nabla^2\phi(x')]^{-1}} \leq M_2(\widetilde{W}_{2,\phi}(\mu,\mu') + \|\nabla\phi(x) - \nabla\phi(x')\|_{[\nabla^2\phi(x')]^{-1}}).$$

In this assumption, we use

$$\widetilde{W}_{2,\phi}^2(\mu,\mu') := \inf_{\pi \in \Pi(\mu,\mu')} \int \|x - x'\|_{[\nabla^2 \phi(x')]^{-1}}^2 d\pi(x, x').$$

This assumption parallels Assumption 2 in the standard Euclidean setting and ensures the existence of the minimizer.[3] Note also that these correspond to the mean-field generalizations of Assumptions 3, 6 from Jiang (2021). For instance, Ahn & Chewi (2021, Section E.2) show that the quadratic potential on the simplex and $\phi(x) = -\sum_i \log x_i$ satisfies this condition in the linear setting, which can be straightforwardly generalized to the mean-field setting.

It is then straightforward to see that (10) admits a unique minimizer; see Appendix C.3 for a proof.

**Theorem 3.1.** *Under Assumptions 3 and 5, (10) is well-posed and admits a unique minimizer $\mu_*$ which satisfies $\mu_* \propto \exp\left(-\frac{1}{\lambda}\frac{\delta F(\mu_*)}{\delta \mu}\right)$.*

Recall that convergence analysis of the mean-field Langevin dynamics utilizes the entropy sandwich inequality (Nitanda et al., 2022; Chizat, 2022). We will also require this result. Remarkably, the statement will hold exactly in the constrained setting as well; please refer to Lemma C.2.

Now we can prove the convergence of the mirror mean-field Langevin dynamics in continuous time. The proof, which mirrors that of ordinary MFLD (Nitanda et al., 2022), is given in Appendix C.5.

**Theorem 3.2.** *Let $\{\mu_t\}_{t \geq 0}$ be the evolution described by (12). Under Assumptions 3– 5,[4] for all $t \geq 0$, it holds that*

$$\mathcal{L}(\mu_t) - \mathcal{L}(\mu_*) \leq e^{-2C_{\mathsf{LSI}}\lambda t}(\mathcal{L}(\mu_0) - \mathcal{L}(\mu_*)).$$

## 4 DISCRETIZATION ANALYSIS FOR MMFLD

### 4.1 OBTAINING THE DISCRETIZED MMFLD

We now derive the time- and space- (i.e. particle-) discretized version of MMFLD by motivating the empirical dynamics from an alternative objective. Indeed, the KL term in (6) no longer makes sense in the finite-particle setting, as the negative entropy is not well-defined for discrete measures. Instead, we study the empirical system by lifting to the configuration space. Let $\mu^{(N)} \in \mathcal{P}^{(N)}$ be a distribution of $N$ particles $\mathbf{X} = (X^i)_{i=1}^N \in \mathcal{X}^N$, where $\mathcal{P}^{(N)}$ is the space of probability measures on $(\mathcal{X}^N, \mathcal{B}(\mathcal{X})^{\otimes N})$. We introduce the following objective on $\mathcal{P}^{(N)}$:

$$\mathcal{L}^{(N)}(\mu^{(N)}) := N \mathop{\mathbb{E}}_{\mathbf{X} \sim \mu^{(N)}}[F(\mu_{\mathbf{X}})] + \lambda \mathsf{Ent}(\mu^{(N)}). \tag{13}$$

We can easily verify that if $\mu^{(N)}$ is the $N$-fold product measure of $\mu \in \mathcal{P}_2(\mathcal{X})$, then $\mathcal{L}^{(N)}(\mu^{(N)}) \geq N\mathcal{L}(\mu)$ by the convexity of $F$ (Suzuki et al., 2023b; Nitanda, 2024), and moreover the optimum of $\mathcal{L}^{(N)}$ is given by the following Gibbs distribution $\mu_*^{(N)}$:

$$\frac{d\mu_*^{(N)}}{d\mathbf{x}}(\mathbf{x}) \propto \exp\left(-\frac{N}{\lambda}F(\mathbf{x})\right).$$

To solve (13), consider the finite-particle approximation of (11), which is described by the system of SDEs $\{\mathbf{X}_t\}_{t \geq 0} = \{(X_t^1, \ldots, X_t^N)\}_{t \geq 0}$:

$$dY_t^i = -\nabla\frac{\delta F(\mu_t)}{\delta \mu}(X_t^i)dt + \sqrt{2\lambda\nabla^2\phi(X_t^i)}dB_t^i \tag{14}$$

with $X_t^i = \nabla\phi^*(Y_t^i)$. We sometimes denote $F(\mathbf{x}) = F(\mu_{\mathbf{X}})$ when emphasizing $F$ as a function of $\mathbf{x}$. Noticing that $N\nabla_{x^i}F(\mathbf{X}_t) = \nabla\frac{\delta F(\mu_{\mathbf{X}})}{\delta \mu}(x^i)$ (Chizat, 2022), we can identify the system (14) with

$$\begin{cases} \mathbf{X}_t &= \nabla\phi^*(\mathbf{Y}_t), \\ d\mathbf{Y}_t &= -N\nabla F(\mathbf{X}_t)dt + \sqrt{2\lambda[\nabla^2\phi^*(\mathbf{X}_t)]^{-1}}d\mathbf{B}_t, \end{cases}$$

---

[3]Note that although this object is not a distance, e.g. it is not symmetric, it is a quadratic approximation of the Bregman divergence associated to $\varphi^*$ at $x'$.

[4]We remark that Assumption 4 should here be the uniform *mirror* LSI, but omit the full definition for space.

---

**Algorithm 1** Discretized MMFLD

---

**Require:** mirror map $\nabla\phi$, timestep $\eta_k$, max iterations $T$, number of particles $N$
1: Initialize $X_0^1, \cdots, X_0^N \sim \mu_0$
2: **for** $k = 0, \cdots, T-1$ **do**
3:      **for** $i \in [N]$ **do**
4:          $Y_0^i \leftarrow \nabla\phi(X_k^i) - \eta_k \nabla\frac{\delta F(\mu_k)}{\delta\mu}(X_k^i)$
5:          Use an Euler-Maruyama discretization to simulate the following diffusion for $t \in [0, \eta_k]$:

$$dY_t^i = \sqrt{2\lambda\left[\nabla^2\phi^*(Y_t^i)\right]^{-1}}\, dB_t$$

6:          $X_{k+1}^i \leftarrow \nabla\phi^*(Y_{\eta_k}^i)$
7:      **end for**
8:      $\mu_{k+1} \leftarrow \frac{1}{N}\sum_{i=1}^N \delta_{X_{k+1}^i}$
9: **end for**
10: **return** $X_T^1, \ldots, X_T^N$

---

where $\mathbf{B}_t$ is the standard Brownian motion on $\mathbb{R}^{dN}$, for sampling from the Gibbs distribution $\mu_*^{(N)}$.

For time discretization of (14), we use the sequence of learning rates $\{\eta_k\}_{k\in\mathbb{N}}$ at each iteration and implement the forward discretization scheme from Ahn & Chewi (2021). Let $\mathbf{X}_k = (X_k^i)_{i=1}^N \subseteq \mathcal{X}$ denote the $N$ particles at the $k$th update, and define $\mu_k = \mu_{\mathbf{X}_k}$ as the corresponding empirical distribution. Starting from $X_0^i \sim \mu_0$, the discretized MMFLD updates $\mathbf{X}_k$ as in Algorithm 1.

The dual algorithm is characterized as follows. Letting $\tilde\mu_0 := \frac{1}{N}\sum_{j=1}^N \delta_{Y_0^i}$, we know that $Y_{k+1}^i = \nabla\phi(X_{k+1})$ is the value at time $t = \eta_k$ of the stochastic process (written in differential form)

$$dY_t^i = -\nabla\frac{\delta F((\nabla\phi^*)_\sharp\tilde\mu_0)}{\delta\mu}(\nabla\phi^*(Y_0^i))dt + \sqrt{2\lambda[\nabla^2\phi^*(Y_t^i)]^{-1}}dB_t^i \tag{15}$$

with initial value given by the previous iteration $\nabla\phi(X_k^i)$.

The forward discretization for mirror Langevin dynamics is known to have vanishing bias as $\eta_k \to 0$, while the standard Euler-Maruyama discretization does not (Ahn & Chewi, 2021; Jiang, 2021) without stronger assumptions on the mirror map $\phi$ (Li et al., 2022). In particular, Li et al. (2022) utilized a mean-square analysis (Li et al., 2019; 2021) to show that under a *modified* self-concordance assumption, the standard Euler-Maruyama discretization does have vanishing bias under the weaker notion of Wasserstein error. We remark that the forward discretization from Ahn & Chewi (2021) (and thus also ours) can be viewed as discretizing the drift but not the diffusion. Ahn & Chewi (2021); Jiang (2021) analyze the discretization where the corresponding SDE in step 5 of Algorithm 1 is simulated exactly. We will follow their approach for simplicity of exposition as the bottleneck of the algorithm is the oracle complexity (number of queries to $\nabla\frac{\delta F}{\delta\mu}$), not tracking the pure diffusion. Nevertheless, our experiments (Section 5) indicate that a one-step discretization generally suffices.

### 4.2 Propagation of Chaos for Mean-Field Networks

For our convergence analysis, we focus on the supervised risk minimization problem of mean-field neural networks (Nitanda et al., 2022; Suzuki et al., 2023b). Let $\mathcal{D}$ be the data space. Let $h(x, \cdot) : \mathcal{D} \to \mathbb{R}$ be a function parameterized by $x \in \mathcal{X}$, representing a single neuron with weight $x$. The mean-field model is obtained by integrating $h(x, \cdot)$ with respect to a probability measure $\mu \in \mathcal{P}_2(\mathcal{X})$ over the parameter space, corresponding to the average output of infinitely many neurons distributed according to $\mu$: $h_\mu(\cdot) = \int h(x, \cdot)d\mu(x)$. Given training examples $\{(z_j, y_j)\}_{j=1}^n \subset \mathcal{D} \times \mathbb{R}$ and loss function $\ell(\cdot, \cdot) : \mathbb{R} \times \mathbb{R} \to \mathbb{R}$, the empirical risk of the mean-field neural network is

$$F(\mu) = \frac{1}{n}\sum_{j=1}^n \ell(h_\mu(z_j), y_j). \tag{16}$$

We use the following assumption from Nitanda (2024):

**Assumption 6.** $\ell(\cdot, y)$ *is convex and smooth, and* $h(X, z)$ *has a finite second moment, e.g.,*

(i) *There exists $L > 0$ such that for every $a, b, y \in \mathbb{R}$, it holds that $\ell(b, y) \leq \ell(a, y) + \frac{\partial \ell(a,y)}{\partial a}(b - a) + \frac{L}{2}|b - a|^2$.*

(ii) *There exists $R > 0$ such that for every $z \in \mathcal{D}$, it holds that $\mathbb{E}_{X \sim \mu_*}|h(X, z)|^2 \leq R^2$.*

This assumption can be verified for many natural settings, including neural networks with bounded activation functions and the logistic and squared losses (Nitanda et al., 2022; Chizat, 2022; Suzuki et al., 2023b; Nitanda, 2024). Using this assumption, we have the following LSI constant-free particle approximation error concerning the objective gap.

**Theorem 4.1.** *Under Assumptions 5 and 6, it holds that*

$$\frac{\lambda}{N}\mathsf{KL}(\mu_*^{(N)} \,||\, \mu_*^{\otimes N}) \leq \frac{1}{N}\mathcal{L}^{(N)}(\mu_*^{(N)}) - \mathcal{L}(\mu_*) \leq \frac{LR^2}{2N}.$$

*Proof.* The arguments in Nitanda (2024, Appendix A) directly hold over $\mathcal{P}_2(\mathcal{X})$ by convexity of $\mathcal{X}$. Note that Nitanda (2024) assumes an additional $L^2$ regularization term $\lambda' \int \|\cdot\|^2 d\mu$, but the proof for the approximation error does not require this. $\qquad\square$

## 4.3 Convergence Analysis for Discretized MMFLD

To utilize the one-step interpolation argument (Vempala & Wibisono, 2019) in our setting, we follow Wibisono (2019); Jiang (2021) and consider the discretized MMFLD as a *weighted* dynamics (see Lemma C.3). The final ingredients we require are self-concordance and the uniform-in-$N$ mirror LSI.

**Assumption 7** (Self-concordance (Gatmiry & Vempala, 2022)). *the conjugate mirror map $\phi^*$ satisfies the following: for every $x \in \operatorname{int} \mathcal{X}$ and $u \in \mathbb{R}^d$, it holds that*

$$|\nabla^3 \phi^*(x)[u, u, u]| \leq 2c_1 \langle u, \nabla^2\phi(x)u\rangle^{3/2}.$$

*Additionally, we let $c_2 := \lambda_{\min}(\nabla^2\phi) > 0$.*

Self-concordance is a standard assumption for interior point and mirror analyses (Nesterov & Nemirovskii, 1994; Bubeck, 2015) and holds for classic examples such as the logarithmic barrier function on a polytope (Gatmiry & Vempala, 2022), hyperbolic-type barriers on convex sets defined by hyperbolic constraints (Narayanan, 2016), or epigraphs of matrix norms (Nesterov & Nemirovskii, 1994). Existing barriers can also be easily combined to provide explicit barriers for (e.g.) product spaces. Moreover, for any $d$-dimensional convex set, there always exists a self-concordant barrier with parameter $O(d)$ (Narayanan, 2016).

**Assumption 8** (Uniform-in-$N$ mirror LSI). *There exists a constant $C_{\mathsf{LSI}} > 0$ such that $\mu_*^{(N)}$ satisfies a mirror log-Sobolev inequality with constant $C_{\mathsf{LSI}}$, that is for every $\mu^{(N)}$, it holds that $\mathsf{KL}(\mu^{(N)} \,||\, \mu_*^{(N)}) \leq \frac{1}{2C_{\mathsf{LSI}}}\mathsf{FI}_\phi(\mu^{(N)} \,||\, \mu_*^{(N)})$.*

This assumption can be verified using Proposition B.1, for instance, via the uniform-in-$N$ LSI (Chewi et al., 2024) and a strongly-convex mirror map.

With these tools, we can finally provide the following convergence guarantee for the discretized MMFLD. The proof is given in Appendix C.6.

**Theorem 4.2.** *Suppose Assumptions 5-8 hold. Then Algorithm 1 with step size $\eta$ satisfies*

$$\frac{1}{N}\mathcal{L}^{(N)}(\mu_k^{(N)}) - \mathcal{L}(\mu_*) \leq \exp\left(-C_{\mathsf{LSI}}\lambda\eta k\right)\left(\frac{1}{N}\mathcal{L}^{(N)}(\mu_0^{(N)}) - \frac{1}{N}\mathcal{L}^{(N)}(\mu_*^{(N)})\right) + \frac{LR^2}{2N} + \frac{\delta_\eta}{2C_{\mathsf{LSI}}\lambda},$$

*where $\delta_\eta := 2\eta M_2^4 M(\eta M_1^2 + 2\lambda d)$, $D := \max_{u,v}\|\nabla\phi(u) - \nabla\phi(v)\|_2$, and $M := \exp\left(\frac{2c_1 D}{\sqrt{c_2}}\right)$.*

This extends the discretization analysis of Nitanda (2024); Nitanda et al. (2025), which to the best of our knowledge constitute the sharpest known bounds for propagation of chaos of MFLD, to the mirror Langevin setting.

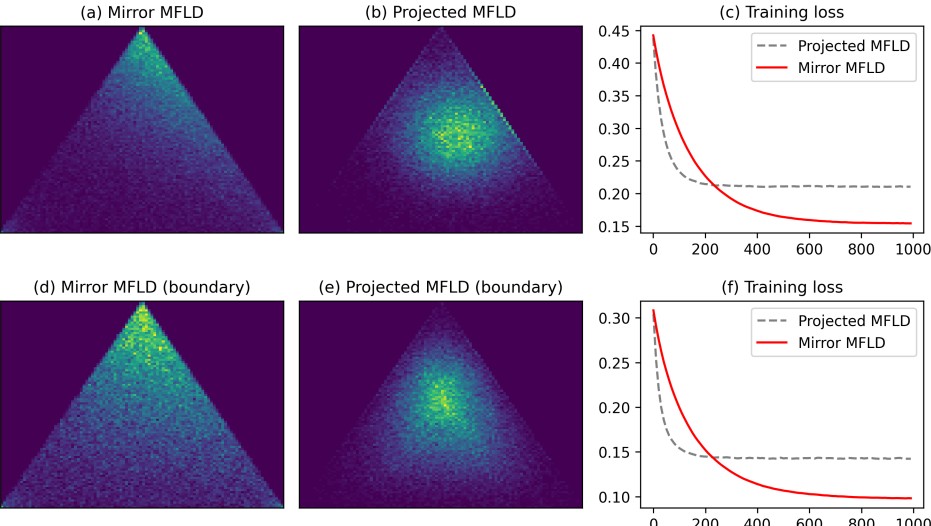

Figure 1: (a)-(c) MMFLD v.s. projected MFLD for optimizing a nonlinear mean-matching objective over the unit simplex; (d)-(f) with an additional boundary potential preventing mass accumulation.

## 5 NUMERICAL EXPERIMENTS

In this section, we conduct experiments demonstrating the utility of (discretized) MMFLD.

### 5.1 OPTIMIZATION OVER THE SIMPLEX

We consider optimization on the unit simplex $\Delta_d = \{x \in \mathbb{R}^d \mid \sum_{i=1}^d x_i = 1, \ x_i \geq 0\}$. A natural choice for the mirror map is the entropy function $\phi(x) = \sum_i x_i \log x_i$, whose dual is given by $\phi^*(y) = \log \sum_i \exp y_i$; see Beck & Teboulle (2003) for details. We set $d = 3$ and aim to minimize the nonlinear functional

$$F(\mu) = \left\| \int_{\Delta_3} x d\mu(x) - q \right\|^2 + \beta \int_{\Delta_3} \sum_{i=1}^3 \log \frac{1}{x_i} d\mu(x).$$

The first term is a mean-matching score for a fixed target $q \in \mathbb{R}^3$, while the second term is a potential barrier preventing collapse towards the boundary $\partial \Delta_3$. As a baseline, we also implement projected MFLD, where particles are projected back onto the simplex after each update (8). The diffusion step of MMFLD is simulated in one step, resulting in similar runtimes. Both algorithms are run using 50K particles with $\eta = 3 \times 10^{-3}$, $\lambda = 0.1$.

Figure 1a-c show the loss curves and final particle distributions when $\beta = 0$. In particular, MFLD exhibits accumulation of mass along the boundary characteristic of projection methods (Figure 1b), which leads to excess error or overfitting. To avoid this trivial failure case, we also add a small barrier preventing convergence to the boundary by setting $\beta = 10^{-4}$ (Figure 1d-f). We observe a drastic effect on projected MFLD as particles are repelled from the boundary; in contrast, MMFLD maintains a more even distribution as the mirror map already ensures the particles are well-behaved near the boundary. In both experiments, MFLD initially converges faster but MMFLD is able to achieve significantly smaller loss, thus demonstrating higher stability and optimality.

### 5.2 OPTIMIZATION OVER THE SPECTRAPLEX

We next study optimization over the *spectraplex* $\mathcal{X} = \{\Sigma \in \mathbb{S}_+^d \mid \mathsf{Tr}(\Sigma) = 1\}$, where $\mathbb{S}_+^d$ is the set of $d \times d$ positive semidefinite matrices. Let $\Sigma^* \in \mathcal{X}$ be a target density matrix. Define the following mean-field functional:

$$F(\mu) = \frac{1}{2} \left\| \mathbb{E}_\mu[\Sigma] - \Sigma^* \right\|_{\mathsf{F}}^2 + \frac{\gamma}{2} \int \|\Sigma\|_{\mathsf{F}}^2 \, d\mu(\Sigma), \quad \gamma > 0.$$

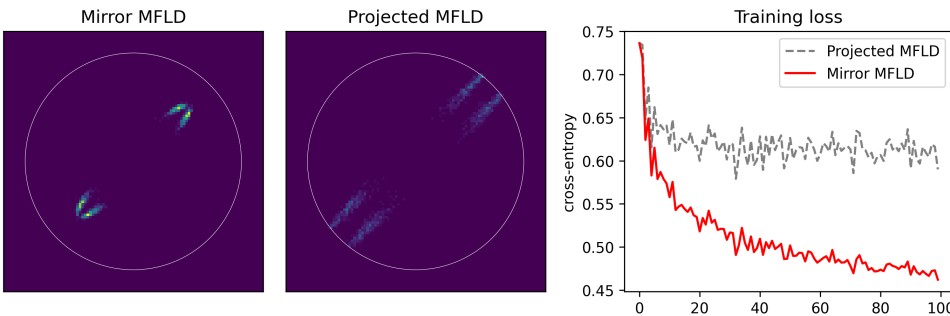

Figure 3: (Left) MMFLD v.s. projected MFLD for optimizing a norm-constrained two-layer ReLU neural network for classification; (right) neuron distribution along XOR directions.

For projected MFLD, the projection step amounts to taking the eigendecomposition, and renormalizing the eigenvalues to sum to 1. For MMFLD, we use the von Neumann mirror map $\phi(\Sigma) = \mathsf{Tr}(\Sigma \log \Sigma - \Sigma)$.

In the experiment, we take $d = 10$, $\gamma = 0.02$, $\eta = 0.1$, $\lambda = 0.1$, and $N = 2048$, and choose a $\Sigma^* \sim \mathrm{Unif}(\mathcal{X})$. Again, the diffusion is simulated using one step. We present the results in Figure 2; we plot $F(\mu)$ as it is easily computed. Observe that projected MFLD makes very little progress, compared to MMFLD. We hypothesize this is due to the projection step roughly canceling out all of the progress we make in the iteration.

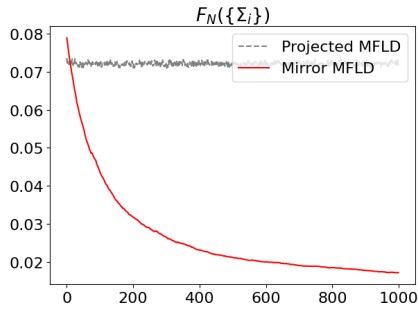

Figure 2: Optimization over the spectraplex with von Neumann mirror map.

### 5.3 CLASSIFICATION WITH MEAN-FIELD NETWORK

We consider the more practical problem of optimizing a norm-constrained neural network for a classification task. Norm constraints are used to stabilize training and improve generalization by preventing overly large parameter values or overfitting. The domain is the $d$-dimensional unit ball with ball barrier $\phi(z) \propto -\log(1 - \|z\|_2^2)$. For the score function, we use a two-layer ReLU network $f(x) = \frac{1}{N} \sum_{i=1}^{N} \mathrm{relu}(\langle w_i, x \rangle)$ with the constraint $\|w_i\|_2 \leq 1$. The prediction is given by the sigmoid transformation $\sigma(f(x))$ with cross-entropy loss. We use XOR data with Gaussian noise, so that the nonlinearity is necessary to learn the decision boundaries. We implement MMFLD and projected MFLD to learn $\{w_i\}_{i=1}^{N}$. Hyperparameters are set as follows: $N = 512$, $d = 2$, $\eta = 0.1$, $\lambda = 10^{-3}$, $T = 100$. Again, the diffusion is simulated in one step.

We plot loss curves and visualize the neuron distribution in the XOR directions in Figure 3. MMFLD decreases the loss at a significantly faster rate than projected MFLD and continues to decrease error after 100 epochs, while projected MFLD stagnates after around 30-50 epochs. The same behavior is observed for the test curves. Moreover, the density plots show that MMFLD is able to align the neurons tightly in the XOR directions, while MFLD scatters the neurons over a wider region and saturates the norm constraint, increasing uncertainty in the prediction and generalization error.

## 6 CONCLUSION

We introduced the mirror mean-field Langevin dynamics to solve distributional optimization problems constrained to a convex subset $\mathcal{X}$ of $\mathbb{R}^d$. Moreover, we obtained linear convergence rates for the continuous-time flow and also proved uniform-in-time propagation of chaos guarantees for the time- and particle-discretized setting. An important direction for future work is to generalize to the mirror Poincaré setting as in Chewi et al. (2020) since it is more natural from a geometric viewpoint (Chewi, 2023); this will require studying the mean-field Langevin dynamics under a Poincaré inequality as well (Nitanda et al., 2022).

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

## A    RELATED WORK

**MLD and constrained sampling.**    For the problem of constrained sampling, the mirrored Langevin dynamics was first proposed by Hsieh et al. (2018). Then, Zhang et al. (2020) introduced the more general mirror Langevin diffusion with Hessian-dependent covariance and its Monte Carlo discretization, the mirror Langevin algorithm (MLA). In continuous time, Chewi et al. (2020) show exponential convergence of MLD under a mirror Poincaré inequality relative to the modified metric. In particular, this implies convergence of the Newton-Langevin diffusion when the mirror map equals the (strictly convex) potential $f$.

The convergence of the discretized algorithm is more difficult to analyze compared to the ordinary Langevin dynamics. Zhang et al. (2020) showed that the MLA iterates contract to a Wasserstein ball around the target density, but with a potentially non-vanishing bias. Ahn & Chewi (2021) proposed a different discretization of MLD which achieves vanishing bias, but assumed an exact computation of the Hessian Riemannian throughout the diffusion process. On the other hand, Li et al. (2022) showed vanishing bias of MLA under more restrictive assumptions on the mirror map based on Li et al. (2019); and Gatmiry & Vempala (2022) utilized an additional second-order self-concordance assumption (cf. Assumption 7) and extensive Riemannian geometry for an exact discretization analysis on Hessian manifolds. Furthermore, Metropolis-adjusted (high accuracy) variants of MLA have been proposed to obtain algorithms whose stationary distributions are unbiased (Srinivasan et al., 2024a;b).

**MFLD and propagation of chaos.**    Initial mean-field analyses of two-layer neural networks were studied independently by several groups (Mei et al., 2018; Chizat & Bach, 2018; Sirignano & Spiliopoulos, 2020; Hu et al., 2021), showing either weak convergence or linear convergence under restrictive assumptions, e.g. the mean-field interaction is small or the optimization is over-regularized (Eberle et al., 2019). Under the Langevin setting, the linear convergence of the MFLD (under mild conditions) was first shown in Nitanda et al. (2022); Chizat (2022), leveraging the log-Sobolev inequality of the proximal Gibbs measure. Whereas Nitanda et al. (2022) studied the time-discretization and offered a primal-dual analysis, Chizat (2022) studied the annealed dynamics where $\lambda_t \to 0$ in (4).

Following works focused on proving propagation of chaos bounds for the corresponding finite particle dynamics, with the noisy gradient descent of finite-width neural networks as a prominent application. While earlier arguments indicated the possibility of error blowing up exponentially in time (Mei et al., 2018), uniform-in-time guarantees were obtained in the case of super-quadratic regularization (Suzuki et al., 2023a) and for more natural quadratic regularization (Chen et al., 2023; Suzuki et al., 2023b; Kook et al., 2024) via a uniform-in-$N$ LSI (Wang, 2024; Chewi et al., 2024). Furthermore, this error was shown to be independent of the LSI constant in Nitanda (2024) and the corresponding convergence analysis was further refined by Chewi et al. (2024); Nitanda et al. (2025), respectively relaxing quadratic regularization to strongly convex regularization and improving the propagation of chaos results using a directional LSI. Also, improved sample complexity results have been established for certain structured classification problems (Suzuki et al., 2023c; Mousavi-Hosseini et al., 2025).

## B    MIRROR LSI

Here, we provide a simple and general method to establish a mirror LSI.

**Proposition B.1** (Daaloul et al. (2025))**.** *If $\nu \in \mathcal{P}(\mathcal{X})$ satisfies a LSI with constant $C_{\mathsf{LSI}}$ and $\phi$ is $\alpha$-strongly convex, then $\nu$ satisfies a mirror LSI with constant $C_{\mathsf{LSI}}/\alpha$.*

For completeness, we provide the proof from (Daaloul et al., 2025).

*Proof.* We have

$$\mathsf{FI}_\phi(\mu \,\|\, \nu) = \int \left\| \nabla \log \frac{\mu}{\nu} \right\|^2_{[\nabla^2 \phi \circ \nabla \phi^*]} d\nu$$
$$\geq \frac{1}{\alpha} \int \left\| \nabla \log \frac{\mu}{\nu} \right\|^2 d\nu$$

$$\geq \frac{2C_{\mathsf{LSI}}}{\alpha} \mathsf{KL}(\mu \,||\, \nu).$$

This concludes the proof. □

As another example, we provide an explicit family of (one-dimensional) distributions that satisfies a mirror LSI, due to Furioli et al. (2019).

**Proposition B.2** (Furioli et al. (2019, Theorem 3)). *Define the densities*

$$\nu_{m,\lambda} \propto (1-x)^{-1+\frac{1-m}{\lambda}}(1+x)^{-1+\frac{1+m}{\lambda}}$$

*for $\lambda > 0$ and $m \in (-1, 1)$. Let the barrier function be*

$$\phi(x) = \frac{1}{2}[(1+x)\log(1+x) + (1-x)\log(1-x)].$$

*Suppose $1 - \frac{\lambda}{2} > 0$ if $m = 0$ and $1 - \frac{\lambda}{2} \geq |m|$, otherwise. Then, $\nu_{m,\lambda}$ satisfies a mirror LSI with constant*

$$C_{\mathsf{LSI}} = \frac{\lambda}{2}\left(1 - \frac{\lambda}{2} + \sqrt{\left(1 - \frac{\lambda}{2}\right)^2 - m^2}\right)^{-1}.$$

### B.1 PROPERTIES OF THE MIRROR LSI

**Lemma B.3** (Stability under bounded perturbation, Jiang (2021, Lemma 2)). *Suppose $\nu$ satisfies a mirror LSI with constant $C_{\mathsf{LSI}}$, then if $c_1 \leq \frac{d\mu}{d\nu} \leq c_2$, then $\mu$ satisfies a mirror LSI with constant $\frac{c_1}{c_2}C_{\mathsf{LSI}}$.*

We remark that Jiang (2021) made a minor typo and stated the flipped result is the mirror LSI constant. For completeness, we provide the proof from Jiang (2021).

*Proof.* Using the variational principle of entropy, for $\mu \ll \nu$, we have

$$\mathsf{Ent}_\mu(g^2) \leq \left\|\frac{d\mu}{d\nu}\right\|_\infty \mathsf{Ent}_\nu(g^2) \leq c_2 \cdot \mathsf{Ent}_\nu(g^2). \tag{17}$$

Next, we have

$$\frac{2}{C_{\mathsf{LSI}}}\int \|\nabla g(x)\|_{[\nabla^2\phi(x)]^{-1}}^2 d\nu = \frac{2}{C_{\mathsf{LSI}}}\int \|\nabla g(x)\|_{[\nabla^2\phi(x)]^{-1}}^2 \frac{d\nu}{d\mu}d\mu$$
$$\leq \frac{2}{c_1}\int \|\nabla g(x)\|_{[\nabla^2\phi(x)]^{-1}}^2 d\mu. \tag{18}$$

Combining (17) and (18), we have

$$\frac{2c_2}{c_1 C_{\mathsf{LSI}}}\int \|\nabla g(x)\|_{[\nabla^2\phi(x)]^{-1}}^2 d\mu \geq \mathsf{Ent}_\mu(g^2),$$

as desired. □

**Lemma B.4** (Tensorization under separable mirror map). *Let $\{\mu_i\}_{i\in[m]}$ be probability measures on $\mathbb{R}^{d_i}$, $\{\phi_i : \mathbb{R}^{d_i} \to \mathbb{R}\}_{i\in[m]}$ be mirror maps that satisfy the conditions in Section 2, and $\mu_i$ satisfies a mirror LSI (with respect to $\phi_i$) with constant $C_{\mathsf{LSI}}^{(i)}$, for every $i \in [n]$. Define $\mu := \bigotimes_{i=1}^m \mu_i$ and the separable map $\phi(x) = \sum_{i=1}^m \phi_i(x_i)$, which satisfies*

$$\nabla^2\phi(x) = \mathsf{diag}(\nabla^2\phi_1(x_1), \ldots, \nabla^2\phi_m(x_m)).$$

*Then $\mu$ satisfies a mirror LSI (with respect to $\phi$) with constant $\min_{i\in[m]} C_{\mathsf{LSI}}^{(i)}$.*

*Proof.* Let $X_i \sim \mu_i$ be independent for $i \in [m]$. Let $X_{-i}$ denote all coordinates except $i$. For any $f \geq 0$, we have

$$
\begin{aligned}
\mathsf{Ent}_\mu(f) &\leq \sum_{i=1}^m \mathbb{E}\left[\mathsf{Ent}_{\mu_i}(f(X_1, \dots, X_m)) \mid X_{-i}\right] \\
&\leq \sum_{i=1}^m \frac{2}{C_{\mathsf{LSI}}^{(i)}} \mathbb{E}\left[\|\nabla_i f(X_1, \dots, X_m)\|_{[\nabla_i^2 \phi_i(X_i)]^{-1}}^2 \mid X_{-i}\right] \\
&\leq \frac{2}{\min_{i \in [m]} C_{\mathsf{LSI}}^{(i)}} \sum_{i=1}^m \mathbb{E}\left[\|\nabla_i f(X_1, \dots, X_m)\|_{[\nabla_i^2 \phi_i(X_i)]^{-1}}^2 \mid X_{-i}\right] \\
&= \frac{2}{\min_{i \in [m]} C_{\mathsf{LSI}}^{(i)}} \mathbb{E}\left[\|\nabla f(X_1, \dots, X_m)\|_{[\nabla^2 \phi(X_1, \dots, X_m)]^{-1}}^2\right],
\end{aligned}
$$

where the first inequality uses tensorization of entropy, the second uses the mirror LSI, and the equality follows from separability of the mirror map. The claim follows. $\qquad\square$

## C    DEFERRED PROOFS AND RESULTS

### C.1    PROPERTIES OF $\phi$

**Lemma C.1** (Properties of $\phi$). *The following hold:*

(i) $\nabla\phi(\mathcal{X}) = \mathbb{R}^d$, *the gradient map* $\nabla\phi : \mathcal{X} \to \mathbb{R}^d$ *is bijective and* $\nabla^2\phi(x) \succ 0$ *for all* $x \in \mathcal{X}$.

(ii) $\nabla\phi^*(y) = \arg\max_{x \in \mathcal{X}} \langle x, y \rangle - \phi(x)$.

(iii) $\nabla\phi^* = (\nabla\phi)^{-1}$, *so* $\nabla\phi(\nabla\phi^*(y)) = y$ *for all* $y \in \mathbb{R}^d$, *and* $\nabla^2\phi(x) = \nabla^2\phi^*(\nabla\phi(x))^{-1}$ *for all* $x \in \mathcal{X}$.

### C.2    PROOF OF THEOREM 2.1

*Proof.* Using the Fokker-Planck PDE of (5), integration by parts, and Assumption 1, we have

$$
\begin{aligned}
\frac{d}{dt}\mathsf{KL}(\mu_t \,\|\, \mu) &= \int \frac{\partial \mu_t}{\partial t}(x) \log \frac{\mu_t}{\mu}(x) dx \\
&= \lambda \int \nabla \cdot \left(\mu_t(x)[\nabla^2\phi(x)]^{-1}\nabla\log\frac{\mu_t}{\mu}(x)\right)\log\frac{\mu_t}{\mu}(x)dx \\
&= -\lambda \int \left\langle \nabla\log\frac{\mu_t}{\mu}, [\nabla^2\phi]^{-1}\nabla\log\frac{\mu_t}{\mu}\right\rangle d\mu_t \\
&= -\lambda\mathsf{FI}(\mu_t \,\|\, \mu) \\
&\leq -2\lambda C_{\mathsf{LSI}} \cdot \mathsf{KL}(\mu_t \,\|\, \mu).
\end{aligned}
$$

The claim follows from Grönwall's inequality. $\qquad\square$

### C.3    PROOF OF THEOREM 3.1

*Proof.* This follows from an analogous argument to that for the existence and uniqueness of the mirror Langevin dynamics (Zhang et al., 2020; Chewi et al., 2020). Well-posedness of (10) in the dual space has been shown, for instance, via Röckner & Zhang (2021). This implies well-posedness in primal space via Lemma C.1. If (10) admits a minimizer, its uniqueness follows from strict convexity of entropy. The property $\mu_* \propto \exp\left(-\frac{1}{\lambda}\frac{\delta F(\mu_*)}{\delta\mu}\right)$ follows from the first-order optimality condition, which says that $\frac{\delta F(\mu_*)}{\delta\mu} + \lambda\log\mu_*$ is a constant $\mu_*$-almost everywhere. Then due to the entropy term, $\mu_*$ has positive density everywhere. Further details can be found in e.g. Mei et al. (2018); Chizat (2022). $\qquad\square$

## C.4 ENTROPY SANDWICH

**Lemma C.2** (Entropy sandwich). *Under Assumption 2, let $\mu_*$ be the unique minimizer of (10). For $\mu \in \mathcal{P}_2(\mathcal{X})$, let $\hat{\mu}$ be the proximal Gibbs measure associated to $\mu$. The following hold.*

*(i) In the sense of functionals on the space of probability density functions,*

$$\frac{\delta \mathcal{L}(\mu)}{\delta \mu} = \lambda \frac{\delta}{\delta \mu'} \mathsf{KL}(\mu' \,||\, \hat{\mu}) \Big|_{\mu' = \mu} = \lambda \log \frac{\mu}{\hat{\mu}}.$$

*(ii) For any $\mu, \mu' \in \mathcal{P}_2(\mathcal{X})$, it holds*

$$\mathcal{L}(\mu) + \int \frac{\delta \mathcal{L}(\mu)}{\delta \mu} d(\mu' - \mu) + \lambda \mathsf{KL}(\mu' \,||\, \mu) \leq \mathcal{L}(\mu').$$

*(iii) For any $\mu \in \mathcal{P}_2(\mathcal{X})$, it holds*

$$\lambda \mathsf{KL}(\mu \,||\, \mu_*) \leq \mathcal{L}(\mu) - \mathcal{L}(\mu_*) \leq \lambda \mathsf{KL}(\mu \,||\, \hat{\mu}).$$

*Proof.* It is straightforward to see that the proof of Nitanda et al. (2022, Proposition 1) still holds over $\mathcal{P}_2(\mathcal{X})$ by convexity of $\mathcal{X}$. $\qquad\square$

## C.5 PROOF OF THEOREM 3.2

*Proof.* We combine Section C.2 and the entropy sandwich inequality in a straightforward manner. From the Fokker-Planck PDE (12) and applying integration by parts, we have by Assumption 4 and Lemma C.2 that

$$\begin{aligned}
\frac{d}{dt}(\mathcal{L}(\mu_t) - \mathcal{L}(\mu)) &= \int \frac{\delta \mathcal{L}(\mu_t)}{\delta \mu}(x) \frac{\partial \mu_t}{\partial t}(x) dx \\
&= \lambda \int \frac{\delta \mathcal{L}(\mu_t)}{\delta \mu}(x) \nabla \cdot \left( \mu_t(x)[\nabla^2 \phi(x)]^{-1} \nabla \log \frac{\mu_t}{\hat{\mu}_t}(x) \right) dx \\
&= -\lambda \int \left\langle \nabla \frac{\delta \mathcal{L}(\mu_t)}{\delta \mu}, [\nabla^2 \phi]^{-1} \nabla \log \frac{\mu_t}{\hat{\mu}_t} \right\rangle d\mu_t \\
&= -\lambda^2 \int \left\langle \nabla \log \frac{\mu_t}{\hat{\mu}_t}, [\nabla^2 \phi]^{-1} \nabla \log \frac{\mu_t}{\hat{\mu}_t} \right\rangle d\mu_t \\
&= -\lambda^2 \mathsf{FI}(\mu_t \,||\, \hat{\mu}_t) \\
&\leq -2\lambda^2 C_{\mathsf{LSI}} \cdot \mathsf{KL}(\mu_t \,||\, \hat{\mu}_t) \\
&\leq -2\lambda C_{\mathsf{LSI}}(\mathcal{L}(\mu_t) - \mathcal{L}(\mu_*)).
\end{aligned}$$

The claim follows from Grönwall's inequality. $\qquad\square$

## C.6 PROOF OF THEOREM 4.2

Before we prove Theorem 4.2, we introduce the characterization of the discretized MMFLD as a weighted dynamics. Using $X_t^i = \nabla \phi^*(Y_t^i)$ and $\mu_t = (\nabla \phi^*)_\sharp \tilde{\mu}_t = \frac{1}{N} \sum_{j=1}^N \delta_{X_t^i}$, we can show the following.

**Lemma C.3.** (15) *is equivalent to the primal dynamics*

$$dX_t^i = \left( \nabla \cdot G_t^i(X_t^i) - G_t^i(X_t^i) \nabla \frac{\delta F(\mu_t)}{\delta \mu}(X_t^i) + \pi_t^i \right) dt + \sqrt{2\lambda G_t^i(X_t)} dB_t^i, \qquad (19)$$

*with* shifted *covariance $G_t^i$ and drift $\pi_t^i$ defined by*

$$G_t^i(X_t^i) = [\nabla^2 \phi(X_t^i)]^{-1},$$

$$\pi_t^i = G_t^i(X_t^i) \left( \nabla \frac{\delta F(\mu_t)}{\delta \mu}(X_t^i) - \nabla \frac{\delta F(\mu_0)}{\delta \mu}(X_0^i) \right).$$

*Proof.* This follows from Itô's lemma, e.g. see Jiang (2021, Appendix C). □

*Proof of Theorem 4.2.* We combine the analyses of Jiang (2021) and Nitanda (2024). We abuse notation to identify the probability distribution with its density function with respect to the Lebesgue measure. For instance, we denote by $\mu_*^{(N)}(\mathbf{x})$ the density of $\mu_*^{(N)}$.

Denote by $\mu_{0t}(\mathbf{x}_0, \mathbf{x}_t)$ the joint probability distribution of $(\mathbf{X}_0, \mathbf{X}_t)$ for time $t$, and by $\nu_{t|0}$, $\nu_{0|t}$, and $\nu_0$, $\nu_t$ the conditional and marginal distributions. We see that $\nu_0 = \mu_k^{(N)} = \mathrm{Law}(\mathbf{X}_k)$, $\nu_\eta = \mu_{k+1}^{(N)} = \mathrm{Law}(\mathbf{X}_{k+1})$, and

$$\nu_{0t}(\mathbf{x}_0, \mathbf{x}_t) = \nu_0(\mathbf{x}_0)\nu_{t|0}(\mathbf{x}_t|\mathbf{x}_0) = \nu_t(\mathbf{x}_t)\nu_{0|t}(\mathbf{x}_0|\mathbf{x}_t).$$

Then, using $F(\mathbf{x}_0) = F(\mu_{\mathbf{x}_0})$ and the resulting equality

$$N\nabla_{x^i}F(\mathbf{x}_0) = \nabla\frac{\delta F(\mu_{\mathbf{x}_0})}{\delta\mu}(x_0^i),$$

we also correspondingly obtain $G_0(\mathbf{x})$ and $\pi_0(\mathbf{x})$ as the diffusion and drift terms at time $t$ when $\mathbf{x}_t = \mathbf{x}$ with $\mathbf{x}_0$ at time $t = 0$.[5] Then using the Fokker-Planck equation for the conditional density $\nu_{t|0}(\mathbf{x}_t|\mathbf{x}_0)$ (Wibisono, 2019, Lemma 3) and the argument in Jiang (2021), we obtain the Fokker-Planck equation of $\nu_t$:

$$\frac{\partial\nu_t(\mathbf{x})}{\partial t}$$

$$= \int \frac{\partial\nu_{t|0}(\mathbf{x}|\mathbf{x}_0)}{\partial t}\nu_0(\mathbf{x}_0)d\mathbf{x}_0$$

$$= \int [-\lambda\nabla\cdot(\nu_{t|0}(\nabla\cdot G_0(\mathbf{x}) - G_0(\mathbf{x})N\nabla F(\mathbf{x})) + \lambda\langle\nabla^2, \nu_{t|0}G_0(\mathbf{x})\rangle$$

$$\quad - \nabla\cdot(\nu_{t|0}\pi_0(\mathbf{x}))]\nu_0(\mathbf{x}_0)d\mathbf{x}_0$$

$$= \lambda\nabla\cdot\left(\nu_{0|t}\int -(\nu_t(\nabla\cdot G_0(\mathbf{x}) - G_0(\mathbf{x})N\nabla F(\mathbf{x}))) + \nabla\cdot(\nu_t G_0(\mathbf{x}))d\mathbf{x}_0\right)$$

$$\quad - \nabla\cdot\left(\nu_t\int\nu_{0|t}\pi_0(\mathbf{x})d\mathbf{x}_0\right)$$

$$= \lambda\nabla\cdot\left(\nu_{0|t}\int G_0(\mathbf{x})\nabla\nu_t + \nu_t G_0(\mathbf{x})N\nabla F(\mathbf{x})d\mathbf{x}_0\right) - \nabla\cdot\left(\nu_t\int\nu_{0|t}\pi_0(\mathbf{x})d\mathbf{x}_0\right)$$

$$= \lambda\nabla\cdot\left(\nu_{0|t}\int\left(\nu_t G_0(\mathbf{x})\nabla\log\frac{\nu_t}{\mu_*^{(N)}}(\mathbf{x})\right)d\mathbf{x}_0\right) - \nabla\cdot\left(\nu_t\int\nu_{0|t}\pi_0(\mathbf{x})d\mathbf{x}_0\right). \quad (20)$$

Now we can control the decrease of the objective as

$$\frac{d\mathcal{L}^{(N)}}{dt}(\nu_t) = \int\frac{\delta\mathcal{L}^{(N)}(\nu_t)}{\delta\mu^{(N)}}(\mathbf{x})\frac{\partial\nu_t}{\partial t}(\mathbf{x})d\mathbf{x}$$

$$= \lambda\int\frac{\delta\mathcal{L}^{(N)}(\nu_t)}{\delta\mu^{(N)}}(\mathbf{x})\nabla\cdot\left(\nu_{0|t}\int\nu_t G_0(\mathbf{x})\nabla\log\frac{\nu_t}{\mu_*^{(N)}}(\mathbf{x})d\mathbf{x}_0\right)d\mathbf{x}$$

$$\quad - \int\frac{\delta\mathcal{L}^{(N)}(\nu_t)}{\delta\mu^{(N)}}(\mathbf{x})\nabla\cdot\left(\nu_t\int\nu_{0|t}\pi_0(\mathbf{x})d\mathbf{x}_0\right)d\mathbf{x}$$

$$= -\lambda\int\nu_{0|t}\int\left\langle\nabla\frac{\delta\mathcal{L}^{(N)}(\nu_t)}{\delta\mu^{(N)}}, \nu_t G_0\nabla\log\frac{\nu_t}{\mu_*^{(N)}}\right\rangle d\mathbf{x}_0 d\mathbf{x}$$

$$\quad + \int\nu_t\left\langle\nabla\frac{\delta\mathcal{L}^{(N)}(\nu_t)}{\delta\mu^{(N)}}, \int\nu_{0|t}\pi_0 d\mathbf{x}_0\right\rangle d\mathbf{x}$$

$$= -\lambda^2\int\nu_{0|t}\int\left\langle\nabla\log\frac{\nu_t}{\mu_*^{(N)}}, \nu_t G_0\nabla\log\frac{\nu_t}{\mu_*^{(N)}}\right\rangle d\mathbf{x}_0 d\mathbf{x}$$

---

[5]Here, note that $G_0(\mathbf{x}) \in \mathbb{R}^{dN\times dN}$ is the block-diagonal covariance matrix.

$$+ \lambda \int \nu_t \left\langle \nabla \log \frac{\nu_t}{\mu_*^{(N)}}, \int \nu_{0|t} \pi_0 d\mathbf{x}_0 \right\rangle d\mathbf{x}$$

$$= -\lambda^2 \mathbb{E}_{\nu_t} \left\| \nabla \log \frac{\nu_t}{\mu_*^{(N)}} \right\|_G^2 + \lambda \mathbb{E}_{\nu_{0t}} \left\langle \pi, \nabla \log \frac{\nu_t}{\mu_*^{(N)}} \right\rangle$$

$$\leq -\lambda^2 \mathbb{E}_{\nu_t} \left\| \nabla \log \frac{\nu_t}{\mu_*^{(N)}} \right\|_{[\nabla^2 \phi]^{-1}}^2 + \frac{1}{2} \mathbb{E}_{\nu_{0t}} \|\pi\|_{\nabla^2 \phi}^2 + \frac{\lambda^2}{2} \mathbb{E}_{\nu_t} \left\| \nabla \log \frac{\nu_t}{\mu_*^{(N)}} \right\|_{[\nabla^2 \phi]^{-1}}^2$$

$$\leq -\lambda^2 C_{\mathsf{LSI}} \mathsf{KL}(\nu_t \,\|\, \mu_*^{(N)}) + \frac{1}{2} \mathbb{E}_{\nu_{0t}} \|\pi\|_{\nabla^2 \phi}^2$$

$$\leq -\lambda C_{\mathsf{LSI}} \left( \mathcal{L}^{(N)}(\nu_t) - \mathcal{L}^{(N)}(\mu_*^{(N)}) \right) + \frac{1}{2} \mathbb{E}_{\nu_{0t}} \|\pi\|_{\nabla^2 \phi}^2, \tag{21}$$

using (20), integration by parts, Young's inequality, and Lemma C.2.

Next, using Assumptions 5, 7, and (15), for

$$M = \exp\left( \frac{2c_1 D}{\sqrt{c_2}} \right), \quad \xi_t = \frac{1 - \exp(-\frac{1}{16c_1^2 t})}{(1 - c_1(tM_1 + 2\sqrt{td}))^2} + \exp\left( -\frac{1}{16c_1^2 t} + \frac{2c_1 D}{\sqrt{c_2}} \right),$$

we may bound

$$\mathbb{E}_{\nu_{0t}} \|\pi\|_{\nabla^2 \phi}^2 \tag{22}$$

$$= \mathbb{E}_{(\mathbf{x}_0, \mathbf{x}_t) \sim \nu_{0t}} \left[ \sum_{i=1}^N \|\pi^i\|_{\nabla^2 \phi(\mathbf{x}_t^i)}^2 \right]$$

$$\leq M_2^2 \mathbb{E}_{(\mathbf{x}_0, \mathbf{x}_t) \sim \nu_{0t}} \left[ \sum_{i=1}^N \|\nabla \phi(\mathbf{x}_0^i) - \nabla \phi(\mathbf{x}_t^i)\|_{[\nabla^2 \phi(\mathbf{x}_t^i)]^{-1}}^2 \right]$$

$$\leq M_2^2 \mathbb{E}_{(\mathbf{x}_0, \mathbf{x}_t) \sim \nu_{0t}} \left[ \sum_{i=1}^N \left\| \nabla \frac{\delta F(\mu_{\mathbf{x}_0})}{\delta \mu}(\mathbf{x}_0^i) - \nabla \frac{\delta F(\mu_{\mathbf{x}_t})}{\delta \mu}(\mathbf{x}_t^i) \right\|_{[\nabla^2 \phi(\mathbf{x}_t^i)]^{-1}}^2 \right]$$

$$\leq M_2^4 \mathbb{E}_{(\mathbf{x}_0, \mathbf{x}_t) \sim \nu_{0t}} \left[ N W_{2,\phi}^2(\mu_{\mathbf{x}_0}, \mu_{\mathbf{x}_t}) + \sum_{i=1}^N \|\mathbf{x}_0^i - \mathbf{x}_t^i\|_{[\nabla^2 \phi(\mathbf{x}_t^i)]^{-1}}^2 \right]$$

$$\leq 2 M_2^4 \mathbb{E}_{(\mathbf{x}_0, \mathbf{x}_t) \sim \nu_{0t}} \left[ \sum_{i=1}^N \|\mathbf{x}_0^i - \mathbf{x}_t^i\|_{[\nabla^2 \phi(\mathbf{x}_t^i)]^{-1}}^2 \right]$$

$$\leq 2 M_2^4 \mathbb{E}_{(\mathbf{x}_0, \mathbf{x}_t) \sim \nu_{0t}} \left[ \sum_{i=1}^N \left\| -t \nabla \frac{\delta F(\mu_{\mathbf{x}_0})}{\delta \mu}(\mathbf{x}_0^i) + \sqrt{2\lambda} \int_0^t [\nabla^2 \phi(\mathbf{x}_s^i)]^{1/2} dB_s \right\|_{[\nabla^2 \phi(\mathbf{x}_t^i)]^{-1}}^2 \right]$$

$$\leq 4 N t^2 M_1^2 M_2^4 \xi_t + 8\lambda t d N M_2^4 M$$

$$= 4 N M_2^4 (t^2 M_1^2 \xi_t + 2\lambda t d M). \tag{23}$$

Here we have used Assumption 5, Itô's isometry, Jensen's inequality, and Lemma C.4.

Now note that $\xi_t \leq M$ deterministically, so for $t \in [0, \eta]$, we have $\frac{1}{2} \mathbb{E}_{\nu_{0t}} \|\pi\|_{\nabla^2 \phi}^2 \leq N\delta_\eta$, where $\delta_\eta := 2\eta M_2^4 M(\eta M_1^2 + 2\lambda d)$. By combining (21) and (23), we see that

$$\frac{d}{dt} \left( \mathcal{L}^{(N)}(\nu_t) - \mathcal{L}^{(N)}(\mu_*^{(N)}) - \frac{N\delta_\eta}{2C_{\mathsf{LSI}}\lambda} \right) \leq -C_{\mathsf{LSI}}\lambda \left( \mathcal{L}^{(N)}(\nu_t) - \mathcal{L}^{(N)}(\mu_*^{(N)}) - -\frac{N\delta_\eta}{2C_{\mathsf{LSI}}\lambda} \right).$$

Noting that $\nu_\eta = \mu_{k+1}^{(N)}$ and $\nu_0 = \mu_k^{(N)}$, Grönwall's inequality yields

$$\mathcal{L}^{(N)}(\nu_{k+1}) - \mathcal{L}^{(N)}(\mu_*^{(N)}) - \frac{N\delta_\eta}{2C_{\mathsf{LSI}}\lambda} \leq \exp\left( -C_{\mathsf{LSI}}\lambda\eta \right) \left( \mathcal{L}^{(N)}(\nu_k) - \mathcal{L}^{(N)}(\mu_*^{(N)}) - \frac{N\delta_\eta}{2C_{\mathsf{LSI}}\lambda} \right).$$

Iterating yields

$$\mathcal{L}^{(N)}(\nu_k) - \mathcal{L}^{(N)}(\mu_*^{(N)}) - \frac{N\delta_\eta}{2C_{\mathsf{LSI}}\lambda} \leq \exp\left( -C_{\mathsf{LSI}}\lambda\eta k \right) \left( \mathcal{L}^{(N)}(\nu_0) - \mathcal{L}^{(N)}(\mu_*^{(N)}) - \frac{N\delta_\eta}{2C_{\mathsf{LSI}}\lambda} \right),$$

from which the claim follows. $\qquad\square$

## C.7 Additional Results

In this section, we collect additional results used in the proof of Theorem 4.2.

The following result generalizes Lemma 4 of Jiang (2021).

**Lemma C.4.** *Under Assumptions 5 and 7, and consider the updates $X_t^i = \nabla \phi^*(Y_t^i)$ following (15), it holds*

$$\frac{1}{M}[\nabla^2 \phi(X_t^i)]^{-1} \preceq [\nabla^2 \phi(X_t^i)]^{-1} \nabla^2 \phi(X_0^i)[\nabla^2 \phi(X_t^i)] \preceq M[\nabla^2 \phi(X_t^i)]^{-1}$$

*in expectation with respect to the Brownian motion for*

$$M = \frac{1 - \exp(-\frac{1}{16c_1^2 t})}{(1 - c_1(tM_1 + 2\sqrt{td}))^2} + \exp\left(-\frac{1}{16c_1^2 t} + \frac{2c_1 D}{\sqrt{c_2}}\right)$$

*if $t \leq \min\left\{\frac{1}{2c_1 M_1}, \frac{1}{16c_1^2 d}\right\}$, and deterministically bounded by $M = \exp\left(\frac{2c_1 D}{\sqrt{c_2}}\right)$. We use the convention $M = 1$ when $c_1 = 0$ and $D = \infty$.*

*Proof.* By Lemma C.1, this implies from Nesterov & Nemirovskii (1994) that for $\|Y_t^i - Y_0^i\|_{\nabla^2 \phi^*(Y_0^i)} \leq \frac{1}{c_1}$, it holds

$$(1 - c_1\|Y_t^i - Y_0^i\|_{\nabla^2 \phi^*(Y_0^i)})^2 \nabla^2 \phi^*(Y_0^i) \preceq \nabla^2 \phi^*(Y_t^i)$$

$$\preceq \frac{1}{(1 - c_1\|Y_t^i - Y_0^i\|_{\nabla^2 \phi^*(Y_0^i)})^2} \nabla^2 \phi^*(Y_0^i).$$

Since

$$\nabla \phi(X_t^i) - \nabla \phi(X_0^i) = Y_t^i - Y_0^i = -t\nabla \frac{\delta F(\mu_0)}{\delta \mu}(X_0^i) + \sqrt{2\lambda t \nabla^2 \phi(X_0^i)} \cdot Z_0^i$$

we have

$$\|Y_t^i - Y_0^0\|_{\nabla^2 \phi^*(Y_0^i)} = \left\|-t\nabla \frac{\delta F(\mu_0)}{\delta \mu}(X_0^i) + \sqrt{2\lambda t \nabla^2 \phi(X_0^i)} Z_0^i\right\|_{[\nabla^2 \phi(X_0^i)^{-1}]}^2$$

$$= t^2 \left\|\nabla \frac{\delta F(\mu_0)}{\delta \mu}(X_0^i)\right\|_{[\nabla^2 \phi(X_0^i)]^{-1}}^2 + 2\lambda t\|Z_0^i\|_2^2$$

$$\leq t^2 M_1^2 + 2\lambda t\|Z_0^i\|_2^2.$$

Using $\chi^2$ concentration from Laurent & Massart (2000), we have

$$\Pr[\|Z_0^i\|_2^2 \geq (\sqrt{d} + \sqrt{\delta})^2] \leq \exp(-\delta)$$

for $t \leq \min\left\{\frac{1}{2c_1 M_1}, \frac{1}{16c_1^2 d}\right\}$. With probability at least $1 - \exp(-d) \geq 1 - \exp\left(-\frac{1}{16c_1^2 d}\right)$ over the draw of $Z_0^i$, it follows that

$$\|Y_t^i - Y_0^i\|_{\nabla^2 \phi^*(Y_0^i)} \leq tM_1 + 2\sqrt{td} < \frac{1}{c_1}.$$

Thus, we have

$$(1 - c_1(tM_1 + 2\sqrt{\lambda td}))^2 I_d \preceq \nabla^2 \phi(X_0^i)^{1/2}[\nabla^2 \phi(X_t^i)]^{-1}\nabla^2 \phi(X_0)^{1/2}$$

$$\preceq \frac{1}{(1 - c_1(tM_1 + 2\sqrt{\lambda td}))^2} I_d.$$

For the remaining probability $\exp\left(-\frac{1}{16c_1^2 d}\right)$, consider the function

$$g_i(s) := \langle u, \nabla^2 \phi^*(Y_0^i + s(Y_t^i - Y_0^i))u\rangle =: \nabla^2 \phi^*(Y_s^i)[u, u].$$

From self-concordance, we have

$$
\begin{aligned}
|g_i'(s)| &= |\nabla^3 \phi^*(Y_s^i)[u, u, Y_t^i - Y_0^i]| \\
&\le 2c_1 \|Y_t^i - Y_0^i\|_{\nabla^2 \phi^*(Y_s)} \|u\|_{\nabla^2 \phi^*(Y_s^i)} \\
&= 2c_1 \|Y_t^i - Y_0^i\|_{\nabla^2 \phi^*(Y_s)} \cdot g_i(s) \\
&\le \frac{2c_1}{\sqrt{c_2}} \|Y_t^i - Y_0^i\|_2 \cdot g_i(s) \\
&\le \frac{2c_1}{\sqrt{c_2}} D \cdot g_i(s),
\end{aligned}
$$

which implies $|\log g(1) - \log g(0)| \le \frac{2c_1 D}{\sqrt{c_2}}$. Then, we deduce

$$
\exp\left(-\frac{2c_1 D}{\sqrt{c_2}}\right) \nabla^2 \phi^*(Y_0^i) \preceq \nabla^2 \phi^*(Y_t^i) \preceq \exp\left(\frac{2c_1 D}{\sqrt{c_2}}\right) \nabla^2 \phi^*(Y_0^i)
$$

and

$$
\exp\left(-\frac{2c_1 D}{\sqrt{c_2}}\right) I_d \preceq \nabla^2 \phi(X_0^i)^{1/2} [\nabla^2 \phi(X_t^i)]^{-1} \nabla^2 \phi(X_0^i)^{1/2} \preceq \exp\left(\frac{2c_1 D}{\sqrt{c_2}}\right) I_d.
$$

Thus, we deduce that, in expectation with respect to $Z_0^i$, $M$ is upper bounded by

$$
M = \frac{1 - \exp(-\frac{1}{16c_1^2 t})}{(1 - c_1(tM_1 + 2\sqrt{\lambda t d}))^2} + \exp\left(-\frac{1}{16c_1^2 t} + \frac{2c_1 D}{\sqrt{c_2}}\right),
$$

which goes to 1 as $t \to 0$. This concludes the proof. $\qquad\square$