# OpenReview forum: "Mirror Mean-Field Langevin Dynamics"
_ICLR.cc/2026/Conference — Submitted to ICLR 2026_

### Official Review · Reviewer_rL3e · 2025-10-31

**Soundness:** 3
**Presentation:** 2
**Contribution:** 3
**Rating:** 4
**Confidence:** 4

**Summary:**

The paper extends mean-field Langevin dynamics (MFLD) to tackle mean-field optimization problems constrained within a convex subset of $\mathbb{R}^d$. To this end, the authors introduce the mirror mean-field Langevin dynamics (MMFLD), which integrates MFLD into the mirror Langevin framework. They establish linear convergence of the continuous-time MMFLD under a uniform logarithmic Sobolev inequality (LSI) and further prove uniform-in-time propagation-of-chaos results for both its time-discretized and particle-discretized counterparts.

**Strengths:**

* The constrained mean-field optimization problem is important to the machine learning community due to its broad and interesting applications, and this paper presents an effective algorithmic approach to address it.

* The paper provides a comprehensive review of relevant prior work.

* The MMFLD formulation, along with its convergence and discretization analyses, is largely comparable to those in the unconstrained setting and constitutes a relatively straightforward extension.

**Weaknesses:**

* The practical applicability of the proposed method remains unclear due to the strong assumptions and the limited, toy-level empirical results.

* The proposed scheme does not constitute a genuine discretization, as it assumes exact simulation of the Brownian motion. Moreover, the analysis largely builds upon existing results from prior works, such as [Ahn & Chewi, 2021] and [Vempala & Wibisono, 2019], and offers limited novelty.

**Questions:**

* The (mirrored) smoothness assumptions are invoked in several theorems, but the corresponding constants are not explicitly reflected in the statements. For instance, is Assumption 2 required for Theorem 2.3? If so, why do the constants $M_1$ and $M_2$ not appear in the theorem? Similarly, is Assumption 5 necessary for Theorem 3.2? And is it also used in Theorem 4.1?

* I understand that the main contribution of this paper is theoretical. However, prior works on mean-field optimization typically include experiments on training two-layer neural networks. It would be valuable if the authors could provide a similar experiment with constrained parameters, which would greatly enhance the practical relevance and applicability of the paper.

* Regarding the experiment, is $\eta = 3 \times 10^{-3}$ chosen as the optimal stepsize for both methods? Different algorithms may exhibit different sensitivities to the stepsize, so using the same value without tuning may raise concerns about fairness in comparison. In addition, since this is an $N$-particle algorithm, it would be helpful to evaluate its sensitivity to the number of particles—for example, by testing $N \in \{256, 512, 1024, 2048, ...\}.

**I will be happy to raise my score if the authors can address my concerns on the assumptions and experiments.**

---

> ### Author Response · Authors · 2025-11-20
>
> **(Part 1/2)** Thank you for the thorough review and helpful comments! Below are our responses.
>
> **The practical applicability of the proposed method remains unclear due to the strong assumptions and the limited, toy-level empirical results.**
> * We emphasize that the assumptions are not strong compared to the literature; we use standard assumptions in both mirror Langevin (relative convexity, smoothness, self-concordance) and mean-field analyses (LSI, uniform LSI). A wide range of standard mirror maps including log-barrier and entropy barriers satisfy these assumptions [HKRC20,Jia21], see also Tables 1,2 of [ZPFP20] for explicit constants.
> * For the assumptions for mean-field analysis, we can verify a (uniform) LSI for bounded first-variation using the Holley-Stroock perturbation method [HS87,SWN23], or for Lipschitz perturbations using [BGMZ18,CG22]. Moreover, a mirror LSI can be verified if the classical LSI is satisfied and $\phi$ is strongly convex [DLLT25]. We will add more detailed discussion on when these assumptions are satisfied.
> * We have added two new experiments which we believe strengthen the empirical applicability and benefits of MMFLD; see our response to Question 2 below.
>
>
> **The proposed scheme does not constitute a genuine discretization, as it assumes exact simulation of the Brownian motion.**
> * We believe a full discretization analysis should incur an extra $O(\eta/k)$ error term, where $\eta$ is the outer step size and we use $k$ steps to discretize step 5 of Algorithm 1. [DLLT25] proved a squared 2-Wasserstein bound of this order, which is weaker than the bound in KL (or equivalently the loss, by entropy sandwich). We attempted to prove a direct KL bound but it seems out of reach using current techniques; [DLLT25] use a coupling-based argument for their result.
> * We also briefly mention that we use the alternative forward discretization (section 4.3 of [Jia21]) that has vanishing bias as $\eta \to 0$, while the Euler-Maruyama discretization (section 4.2 of [Jia21]) has irreducible bias as $\eta\to 0$. The algorithm would work with either discretization, but we chose the former to obtain convergent bounds.
>
> **The (mirrored) smoothness assumptions are invoked in several theorems, but the corresponding constants are not explicitly reflected in the statements. For instance, is Assumption 2 required for Theorem 2.3? If so, why do the constants and not appear in the theorem? Similarly, is Assumption 5 necessary for Theorem 3.2? And is it also used in Theorem 4.1?**
> * The constants $M_1,M_2$ only show up in the discretization analysis. We chose to keep these in Assumption 2 to mirror that of Assumption 5. Nevertheless, the assumptions are still required for the continuous-time results to ensure the existence of a strong solution to the respective SDEs.
>
> **Prior works on mean-field optimization typically include experiments on training two-layer neural networks. It would be valuable if the authors could provide a similar experiment with constrained parameters, which would greatly enhance the practical relevance and applicability of the paper.**
>
> * We have added **two new experiments** on (1) distributional optimization over the space of symmetric positive semidefinite matrices, and (2) the requested experiment on weight-constrained two-layer ReLU neural networks for nonlinear classification, which further demonstrate the empirical benefits of mirror MFLD in more practical settings. Please see the updated Section 5 for details.
>
> * For (1), we use the von Neumann mirror map $\text{Tr}(\Sigma\log\Sigma - \Sigma)$ on the spectraplex and optimize Frobenius error, while for (2), we study an L2 norm constraint via the ball barrier $-\log(1-\lVert z\rVert_2^2)$ and optimize logistic loss against noised XOR data. Hence, the new experiments cover a wide variety of relevant settings for optimization.
>
> * To summarize the results: for (1), we show that MMFLD successfully converges to the target matrix while projected MFLD barely makes any progress. For (2), we demonstrate that MMFLD decreases the loss at a significantly faster rate than projected MFLD, which stagnates quickly. Projected MFLD also tends to saturate the norm constraint, leading to overfitting and increased generalization gap/test error.
>
> (continued)

---

> ### Author Response · Authors · 2025-11-20
>
> **(Part 2/2)**
>
> **Regarding the experiment, is $\eta = 3\times 10^{-3}$ chosen as the optimal stepsize for both methods? Different algorithms may exhibit different sensitivities to the stepsize, so using the same value without tuning may raise concerns about fairness in comparison. In addition, since this is an $N$-particle algorithm, it would be helpful to evaluate its sensitivity to the number of particles—for example, by testing $N \in \{256, 512, 1024, 2048, ...\}$.**
>
> * The $\eta$ values of both algorithms can be increased to $0.01$ (which just results in both curves being scaled by the same factor). When $\eta$ is larger than around $0.015$, both algorithms start to fail; for MMFLD, particles start to cluster at the vertices due to overflow; for projected MFLD, all particles concentrate on the boundary. For the updated experiments, we also find that tuning $\eta$ does not help projected MFLD, while decreasing $\eta$ to be less than $0.1$ for mirror MFLD just slows down optimization.
>
>
> * We have also repeated the experiments with different values of $N$. Decreasing $N$ (e.g., less than $128$) just tends to make loss curves more unstable, while increasing $N$ beyond $1024$ does not really change the results. Indeed, this is expected from the theoretical analysis since $N$ only affects the discretization error upper bound, not convergence speed.
>
> We sincerely thank the reviewer for their helpful comments, and humbly ask to consider raising their score if major concerns have been satisfactorily addressed.
>
>
> [DLLT25] Daaloul et al., 2025. Convergence of the mirror Langevin algorithm on unbounded domains under log-Sobolev inequality for the target measure. https://hal.science/hal-05127974v1
>
> [BGMZ18] J.-B. Bardet, N. Gozlan, F. Malrieu, and P.-A. Zitt. Functional inequalities for Gaussian convolutions of compactly supported measures: Explicit bounds and dimension dependence. Bernoulli, 24(1):333–353, 2018.
>
> [CG22] P. Cattiaux and A. Guillin. Functional inequalities for perturbed measures with applications to log-concave measures and to some Bayesian problems. Bernoulli, 28(4):2294–2321, 2022.

---

### Official Review · Reviewer_qe14 · 2025-10-31

**Soundness:** 3
**Presentation:** 3
**Contribution:** 2
**Rating:** 2
**Confidence:** 4

**Summary:**

This paper studies an extension of mean-field Langevin dynamics to constrained settings using ideas from mirror Langevin dynamics. They obtain linear convergence of continuous time dynamics under a uniform log-Sobolev inequality, and also give a propagation of chaos result to give a result that is discrete in space and time. A numerical experiment on the simplex is given that shows some slight advantage to the mirror mean-field Langevin dynamics.

**Strengths:**

- The paper is generally well written and easy to understand. The theorem statements are generally clear to me and the proofs are readable and easy to follow.
- This paper represents a natural extension of mean-field Langevin dynamics using a mirror map. As was done in the non-mean-field case, this extension has desirable properties of relying on relative Lipschitz types of assumptions, and also naturally maintains the constraints of the problem.
- The results utilize the full range of available tools to prove a convergence bound for a particle distribution in discrete time.

**Weaknesses:**

- The theorems given seem to be standard extensions of existing results in the literature. The heavy lifting seems to have been done in past works like Nitanda et al '22, Nitanda '24, and Nitanda et al '25. Because of this, I am worried that this work is more of a synthesis work than giving some novel and new ideas that would be sufficient for publication in ICLR.
- Coupled with the above limited theoretical novelty, there is a lack of experimental evidence. The authors only give one low-dimensional experiment that barely shows an advantage for MMFLD.
- There is little attention paid to a motivating example. Why should the reader care about constrained sampling? Grounding this problem in real problems of interest to the machine learning community would greatly strengthen the position of this paper.

**Questions:**

- For the continuous setting, in Assumption 4 the authors assume a uniform LSI. It would be useful if the authors could provide a discussion of the cases when this holds rather than offloading this to references. The same comment goes for Assumptions 6 and 8.
- The modified Wasserstein distance is not symmetric in $\mu, \mu'$? Does this have any connnection to a Bregman Wasserstein divergence, or some other "distance" of interest?
- The constant in Theorem 4.2 is exponential in $D$, and for common barrier mirror maps, $\nabla \phi$ is surjective. Doesn't this mean that $D$ is unbounded? The authors should comment on this.
- Where do I see the self concordance parameter $\gamma_1$ in the statement of Theorem 4.2?
- Can the authors comment on their results in comparison with the analysis of projected Langevin (Bubeck et al '15), projected SGLD (Lamperski '21), and other recent methods for sampling from convex bodies (like Gu et al '24)? Having a more in depth theoretical comparison of the advantages of a mirror approach in this setting would be useful, and could also help to guide experimental settings to show where this method has a true advantage.

References:
Bubeck, Sebastien, Ronen Eldan, and Joseph Lehec. "Finite-time analysis of projected Langevin Monte Carlo." Advances in Neural Information Processing Systems 28 (2015).
Lamperski, Andrew. "Projected stochastic gradient langevin algorithms for constrained sampling and non-convex learning." Conference on Learning Theory. PMLR, 2021.
Gu, Yuzhou, et al. "Log-concave sampling from a convex body with a barrier: a robust and unified dikin walk." Advances in Neural Information Processing Systems 37 (2024): 69230-69298.

---

> ### Author Response · Authors · 2025-11-20
>
> **(Part 1/2)** Thank you for the careful reading of our paper and detailed comments. Below are our responses.
>
> **Weaknesses:**
>
> **The theorems given seem to be standard extensions of existing results in the literature.**
>
> * While our result combines existing techniques for MFLD and mirror dynamics, it also represents the first time MFLD has been extended to boundary-constrained domains with provable convergence guarantees, while existing works have mostly focused on $\mathbb{R}^d$ or the torus. In particular, the proposed algorithm is not what is usually run for such problems (projection steps or Riemannian MFLD), and thus can also hopefully inform practitioners on more stable and high-performance optimization methods. Moreover, the LSI-independent analysis of uniform-in-time propagation of chaos [NLK+25] is quite new and we believe it is important to expand on possible extensions of the theory, rather than trying to come up with completely new proof approaches for every submission. Hence we firmly believe our contribution is of value to the community beyond technical novelty.
>
> **The authors only give one low-dimensional experiment that barely shows an advantage for MMFLD.**
>
> * We have added **two new experiments** on (1) distributional optimization over the space of symmetric positive semidefinite matrices, and (2) weight-constrained two-layer ReLU neural networks for nonlinear classification, which further demonstrate the empirical benefits of mirror MFLD in more practical settings. Please see the updated Section 5 for details.
>
> * For (1), we use the von Neumann mirror map $\text{Tr}(\Sigma\log\Sigma - \Sigma)$ on the spectraplex and optimize Frobenius error, while for (2), we study an L2 norm constraint via the ball barrier $-\log(1-\lVert z\rVert_2^2)$ and optimize logistic loss against noised XOR data. Hence, the new experiments cover a wide variety of relevant settings for optimization.
>
> * To summarize the results: for (1), we show that MMFLD successfully converges to the target matrix while projected MFLD barely makes any progress. For (2), we demonstrate that MMFLD decreases the loss at a significantly faster rate than projected MFLD, which stagnates quickly. Projected MFLD also tends to saturate the norm constraint, leading to overfitting and increased generalization gap/test error.
>
> * We also remark that the experiments are low-dimensional because we also aim to visualize the converged distributional solution using a 2D histogram, which gives a much richer picture of the behavior of MMFLD v.s. projected MFLD (Figures 1 and 3). We have also ran the same experiments in higher dimensions and used slicing, but this merely results in lower quality visualizations with essentially the same loss curves, so we elected to include the former. Even such in simple settings, the qualitative and quantitative separation is already quite clear in all three experiments.
>
> **Grounding this problem in real problems of interest to the machine learning community would greatly strengthen the position of this paper.**
>
> * In the introduction, we have given concrete problems of interest which have been extensively studied in many influential works in machine learning and optimization: "many applications of MFLD require the domain of optimization to be (implicitly or explicitly) constrained, e.g., trajectory inference (Chizat et al., 2022; Gu et al., 2025a;b), computation of Wasserstein barycenters (Chizat, 2023; Vaskevicius & Chizat, 2023), computation of discrepancy measures (Suzuki et al., 2023b) or signal deconvolution (Chizat & Bach, 2018) with bounded support, and optimization of neural networks (Nitanda et al., 2022; 2025) with constrained parameters."
>
> * As Reviewer 8nHW mentions, our ideas can also be utilized to obtain novel convergence analyses (e.g., in conjunction with bilevel optimization) for two-layer neural networks, which is a key interest in the theoretical ML community.
>
> * Moreover, we have added the aforementioned new experiments (optimization over PSD matrices, and optimization of neural networks for classification) which further ground our methods in practical problems in optimization. If there are other relevant areas which we missed, please let us know and we will add more in-depth discussion.
>
> (continued)

---

> ### Author Response · Authors · 2025-11-20
>
> **(Part 2/2) Questions:**
>
> **For the continuous setting, in Assumption 4 the authors assume a uniform LSI.**
> * We will add discussion on Assumptions 4,8 in the appendix alongside the verification of mirror LSI. Generally, in the case of bounded first-variation, this is obtained from the classical Holley-Stroock and Bakry-Emery perturbation arguments with the L2 regularization acting as the strongly convex term. For Lipschitz perturbations of strongly convex potential, we can instead use the results in [BGMZ18,CG22]. For Assumption 6, it is clear that logistic and squared loss satisfy (i) and bounded activations satisfy (ii).
>
> **The modified Wasserstein distance is not symmetric in $\mu,\mu'$.**
> * This is correct. In the revision, we will remove "distance" to prevent confusion. It is a quadratic approximation of the Bregman divergence associated with $\varphi^*$ at $x$.
>
> **The constant in Theorem 4.2 is exponential in $D$.**
> * Indeed, the dependence of $M$ on $D$ limits the applicability of Theorem 4.2 to maps with stable Hessian. This is a known issue for the forward discretization, e.g., see [Jia21], Proposition 3 and the discussion thereafter.
>
> **Where do I see the self concordance parameter $\gamma_1$ in the statement of Theorem 4.2?**
> * This is a typo; thanks for catching it. In the theorem $c_1$ is $\gamma_1$ and $c_2$ is $c$. We have fixed this in the revision.
>
> **Can the authors comment on their results in comparison with the analysis of projected Langevin (Bubeck et al '15), projected SGLD (Lamperski '21), and other recent methods for sampling from convex bodies (like Gu et al '24)?**
> * Applying projected Langevin to interacting particle systems, as the experiments suggest, can lead to mass concentrating by the boundary, which is intuitively a larger issue in the mean-field setting compared to the linear setting, e.g. it is much less likely for one particle to be near the boundary vs N particles in MFLD.
> * The rough issue with previous works on sampling from convex bodies is that they sample with respect to a single fixed target, whereas in each iteration of MFLD, the proximal Gibbs distribution evolves. Hence, the analysis must be combined with the mean-field dynamical analysis, which is a new contribution.
>
> We thank the reviewer again for their insightful critiques which have greatly helped to improve the manuscript, and humbly ask the reviewer to consider raising their score if their main concerns, such as lack of motivating examples and experimental evidence, have been addressed with our new empirical results.
>
> **References.**
>
> [NLK+25] Atsushi Nitanda, Anzelle Lee, Damian Tan Xing Kai, Mizuki Sakaguchi, and Taiji Suzuki. Propagation of chaos for mean-field Langevin dynamics and its application to model ensemble. https://arxiv.org/abs/2502.05784
>
> [Jia21] Qijia Jiang. Mirror Langevin Monte Carlo: the case under isoperimetry. In Advances in Neural Information Processing Systems, 2021.
>
> [BGMZ18] J.-B. Bardet, N. Gozlan, F. Malrieu, and P.-A. Zitt. Functional inequalities for Gaussian convolutions of compactly supported measures: Explicit bounds and dimension dependence. Bernoulli, 24(1):333–353, 2018.
>
> [CG22] P. Cattiaux and A. Guillin. Functional inequalities for perturbed measures with applications to log-concave measures and to some Bayesian problems. Bernoulli, 28(4):2294–2321, 2022.

---

> > ### Comment · Reviewer_qe14 · 2025-11-24
> >
> > I thank the authors for their detailed response and the revised manuscript. The additional experiments and clarifications have improved the paper, and some of my earlier concerns (e.g., about motivation and some technical details) are partially mitigated. However, I still don’t believe that this work quite meets the bar of acceptance to ICLR.
> >
> > First, the positives. I see the references to how the formulation explicitly connects to applications such as trajectory inference, Wasserstein barycenters, and constrained neural networks, both in the introduction and in the related work. The new experiments on optimization over the spectraplex using the von Neumann mirror map and on a norm‑constrained two‑layer ReLU network do a better job of illustrating qualitative differences between mirror MFLD and projected MFLD (e.g., boundary mass accumulation and norm saturation). The authors clarified the status of the “modified Wasserstein distance” (now correctly treated as a quadratic approximation of a Bregman divergence), fixed the typo regarding the self‑concordance constants in Theorem 4.2, and added some discussion on how mirror LSIs can be obtained from classical LSIs.
> > That said, several core issues remain.
> > 1. Even in this response and edits, the question of a motivating problem is largely offloaded to other works. It is not at all easy to see how these motivating problems can be formulated using the proposed objective, as well as why this formulation is natural and useful. So the reader is still left to wonder concretely why this is actually useful and why we should be considered MMFLD.
> > 2. I still view the theoretical contribution as incremental. The main conceptual step is to combine mirror Langevin dynamics with mean‑field Langevin dynamics to handle convex constrained domains, leading to the MMFLD SDE and its Fokker–Planck equation. The continuous‑time convergence analysis (Theorem 3.2) is essentially a mirror‑geometry version of the existing MFLD argument: it seems to rely on a similar framework and technique to proofs in the unconstrained setting. Likewise, the discretization and propagation‑of‑chaos analysis (Theorem 4.2) closely follow prior work, with the main technical work being to adapt Lipschitz/smoothness and LSI assumptions to mirror geometry and to track the Hessian metric.
> > 3. Beyond the previous, my main remaining technical concern is that the discrete‑time/finite‑particle convergence theorem (Theorem 4.2), which is presented as a key contribution, appears not to cover the most interesting mirror maps used in the paper and in typical constrained problems. The constant in Theorem 4.2 remains a problem. For classical barrier‑type mirror maps on constrained domains (entropy barrier on the simplex, von Neumann entropy on the spectraplex, ball barrier), the gradient is unbounded near the boundary, so D is infinite and thus the bound in Theorem 4.2 is vacuous. Indeed, don't all three experiments in Section 5 use precisely such barrier mirrors (entropy on the simplex, von Neumann entropy on the spectraplex, ball barrier for the norm‑constrained network)? So the main discrete‑time propagation‑of‑chaos guarantee does not actually apply to the algorithmic setups showcased empirically. The authors acknowledge this issue in the rebuttal by pointing out that this kind of poor constant is a known limitation of forward discretization (cf. Jiang, 2021). This means the discrete‑time theorem is effectively restricted to mirror maps with globally bounded gradient/Hessian, which excludes many of the most natural constrained settings the paper aims to address, such as those in the experiments. This mismatch between theory and practice significantly weakens the claimed contribution.
> > 4. The empirical evaluation, while expanded, is still quite limited. If the authors want to sell what they have as an algorithmic rather than a theoretical contribution, then these would need to be expanded to more of the “real” settings where they claim that this is useful.
> >
> > Given that I don’t view the theoretical contribution as particularly strong and the empirical story as particularly convincing, I am not inclined to raise my score above a 4. I may be convinced if the authors could add explanations of
> > - What specifically the *technical innovations* of the work are,
> > - A discussion of how the proposed theoretical framework is useful for the analysis of constrained MD algorithms, if it cannot extend to barriers, and
> > - If the paper is to be sold as an algorithmic contribution, give a more in-depth explanation of where, in particular, the MMFLD is expected to be used.

---

### Official Review · Reviewer_ZizB · 2025-11-03

**Soundness:** 3
**Presentation:** 4
**Contribution:** 2
**Rating:** 6
**Confidence:** 3

**Summary:**

The authors consider a combination of mirror Langevin dynamics and mean-field Langevin dynamics, analysing the setting of entropy-regularised functionals on constrained convex domains. They provide convergence guarantees for the continuous-time flow under logarithmic Sobolev inequalities and develop guarantees for a time- and particle-discretized scheme. They also provide experiments comparing the scheme to projected mean-field Langevin dynamics, showing that their scheme attains a lower final loss.

**Strengths:**

* The authors target a fairly important and surprisingly open problem, since mean-field dynamics are used to understand two-layer neural networks and mirror descent is frequently used in constrained optimisation.
* The paper is very well-written.
* The guarantees are strong and are under relatively standard conditions in this area (e.g., uniform LSI).

**Weaknesses:**

* The majority of the proof techniques appear to be borrowed or adapted from other papers (e.g., Nitanda et al., 2022; Jiang, 2021 and Nitanda, 2024), so the work may have limited technical novelty at the proof level.
* The analysis of the discretized algorithm assumes that the pure diffusion step (Algorithm 1, step 5) can be **simulated exactly**. The authors note this is for "simplicity of exposition", but this is rarely possible in practice and creates a gap between the theory and the implementation (which used a one-step discretization).
* The discrete-time convergence analysis (Section 4.3) is presented for the specific setting of the mean-field neural network risk minimization problem, which may limit the perceived generality of the result.

**Questions:**

* What are the primary technical novelties of this work at the level of the proof, beyond the synthesis of existing analytical frameworks?
* Regarding Weakness 2: Can the authors comment on the error introduced by *not* simulating the diffusion step exactly? Would a practical, one-step discretization of this diffusion term impact the final convergence guarantee in Theorem 4.2?

---

> ### Author Response · Authors · 2025-11-20
>
> Thank you for the positive assessment of our work and helpful comments! Below are our responses.
>
> **Weakness 1 & Question 1: What are the primary technical novelties of this work at the level of the proof, beyond the synthesis of existing analytical frameworks?**
>
> * While our result combines existing techniques for MFLD and mirror dynamics, it also represents the first time MFLD has been extended to boundary-constrained domains with provable convergence guarantees, while existing works have mostly focused on $\mathbb{R}^d$ or the torus. In particular, the proposed algorithm is not what is usually run for such problems (projection steps or Riemannian MFLD), and thus can also hopefully inform practitioners on more stable and high-performance optimization methods. Moreover, the LSI-independent analysis of uniform-in-time propagation of chaos [NLK+25] is quite new and it is important to expand on possible extensions of the theory. Hence we firmly believe our contribution adds value beyond novelty of proof techniques.
> * We also add experiments on (1) weight-constrained mean-field neural networks for classification, and (2) distributional optimization over the space of symmetric positive semidefinite matrices, which further demonstrate the empirical benefits of mirror MFLD. Please see the updated Section 5.
>
> **Weakness 2 & Question 2: Can the authors comment on the error introduced by not simulating the diffusion step exactly? Would a practical, one-step discretization of this diffusion term impact the final convergence guarantee in Theorem 4.2?**
>
> * We believe the excess discretization error showing up on the right-hand side of Theorem 4.2 should be $O(\eta/k)$, where $\eta$ is the outer step size and we use $k$ steps to discretize step 5 of Algorithm 1. [DLLT25] proved a squared 2-Wasserstein bound of this order, which is weaker than the bound in KL (or equivalently the loss, by entropy sandwich). We attempted to prove a direct KL bound but it seems out of reach using current techniques; [DLLT25] use a coupling-based argument for their result.
> * We also briefly mention that we use the alternative forward discretization (section 4.3 of [Jia21]) that has vanishing bias as $\eta \to 0$, while the Euler-Maruyama discretization (section 4.2 of [Jia21]) has irreducible bias as $\eta\to 0$. The algorithm would work with either discretization, but we chose the former to obtain convergent bounds.
> * In our experiments, we generally found that a one-step discretization is sufficient. Multiple steps tend to only increase runtime while achieving basically the same performance.
>
> **Weakness 3: The discrete-time convergence analysis (Section 4.3) is presented for the specific setting of the mean-field neural network risk minimization problem, which may limit the perceived generality of the result.**
>
> * We decided to present the stronger LSI constant-free propagation of chaos result from [Nit24] for ease of presentation. The weaker result from [CRW23] likely holds more generally. We will check and add this to the revision.
>
>
> [CRW23] Chen, Ren, and Wang. Uniform-in-time propagation of chaos for mean field Langevin dynamics. https://arxiv.org/abs/2212.03050
>
> [Nit24] Nitanda. Improved particle approximation error for mean field neural networks. https://arxiv.org/abs/2405.15767
>
> [NLK+25] Atsushi Nitanda, Anzelle Lee, Damian Tan Xing Kai, Mizuki Sakaguchi, and Taiji Suzuki. Propagation of chaos for mean-field Langevin dynamics and its application to model ensemble. https://arxiv.org/abs/2502.05784
>
> [DLLT25] Daaloul et al. Convergence of the mirror Langevin algorithm on unbounded domains under log-Sobolev inequality for the target measure. https://hal.science/hal-05127974v1

---

### Official Review · Reviewer_8nHW · 2025-11-03

**Soundness:** 3
**Presentation:** 3
**Contribution:** 3
**Rating:** 8
**Confidence:** 4

**Summary:**

This paper introduces and studies mirror mean-field Langevin dynamics, which can be used to solve constrained distributional optimization problems. The authors establish the exponential convergence of this dynamics under a (mirror) log-Sobolev inequality akin to the Euclidean counterpart, and also carry out a complete discretization analysis on both time and the number of particles.

**Strengths:**

This paper provides a clean and novel analysis of mirror mean-field Langevin dynamics as a generalization of mean-field Langevin, and the numerical illustration demonstrates that this can be a better idea to solve distributional optimization problems compared to projected mean-field Langevin. This is interesting in particular since currently there are not many well-studied algorithms for distributional optimization that work well in high dimensions beyond the mean-field Langevin dynamics.

**Weaknesses:**

* I think the authors can better motivate the study of mirror mean-field Langevin by showing what new settings can be unlocked by their analysis, e.g. for training weight-constrained two-layer neural networks or for generative modeling.

* Since the discretization cost of Step 5 is not analyzed, it could potentially be helpful to have, perhaps an informal, discussion of why simulating this step is easier than simulating a Brownian motion on $\mathcal{X}$ (and consequently performing MFLD on $\mathcal{X}$).

**Questions:**

A typical framework for the theoretical study of two-layer nets is to constrain the first layer weights to live on the unit sphere, and allow the second layer weights to be unbounded. A challenge here is that one can not show uniform LSI due to the unbounded weights of the second layer. One way to remedy the issue is to perform bilevel optimization as in [1], which reduces the problem to MFLD on the unit sphere with a bounded uniform-LSI constant. However, [1] still requires performing MFLD on the unit sphere, without discretization guarantees. A concrete application of the results of this paper can be to instead use mirror MFLD after the bilevel reduction, to provide an end-to-end guarantee for training two-layer networks in the mean-field regime.

[1] G. Wang et al, "Mean-Field Langevin Dynamics for Signed Measures via a Bilevel Approach." NeurIPS 2024.

---

> ### Author Response · Authors · 2025-11-20
>
> Thank you for the detailed review and positive assessment of our contributions! Below are our responses.
>
> **The authors can better motivate the study of mirror mean-field Langevin by showing what new settings can be unlocked by their analysis, e.g. for training weight-constrained two-layer neural networks.**
> * We have added additional examples in the revised pdf. In particular, we have added two new experiments on (1) weight-constrained mean-field neural networks for classification, and (2) distributional optimization over the space of symmetric positive semidefinite matrices, which further demonstrate the empirical benefits of mirror MFLD. Please see the updated Section 5.
>
> **Since the discretization cost of Step 5 is not analyzed, it could potentially be helpful to have, perhaps an informal, discussion of why simulating this step is easier than simulating a Brownian motion on $\mathcal{X}$.**
> * Roughly speaking, for common choices of mirror maps, e.g. negative entropy on the simplex, log-barriers, both $\nabla\phi$ and $[\nabla^2\phi]^{-1/2}$ admit closed forms, or are efficiently computable, so the implementation of Algorithm 1 is relatively straightforward. On the other hand, simulating a Brownian motion explicitly on a constrained manifold is substantially harder and manifold-specific, e.g. [LE23]. We will add this in the revision.
>
> **On the bilevel approach.**
> * We thank the reviewer for bringing up the related work. Indeed, bilevel+mirror MFLD could potentially be used to obtain a novel end-to-end theoretical guarantee for networks with unbounded second layer. We leave this as an interesting direction for future work.
>
> [LE23] Li and Erdogdu. Riemannian Langevin algorithm for solving semidefinite programs. arXiv:2010.11176, 2023.

---

### Author Response · Authors · 2025-12-03

Dear Area Chair,

We would like to summarize the paper's status and the main improvements made during the discussion phase. In particular, while the initial scores are varied (2/4/6/8), we believe our rebuttals should have raised them to at least 4/6/6/8; please see below.

**Summary:** There is broad agreement that our paper addresses an important and underexplored problem: mean-field optimization on convex constrained domains, with direct relevance to constrained neural networks and Wasserstein-type problems. Reviewers concur that MMFLD gives a principled way to extend mean-field Langevin dynamics to this setting with rigorous convergence guarantees for both continuous-time and discretized schemes. With the added numerical experiments and visualizations (matrix optimization over the spectraplex, and classification with norm-constrained ReLU networks), the paper now also demonstrates consistent empirical advantages over projected MFLD.

Two reviewers raised concerns centered on perceived incremental theoretical novelty, opaqueness of certain assumptions and narrow scope of experiments. In our revision, we directly addressed these points: we added the aforementioned experiments to strengthen the algorithmic value of MMFLD, clarified when our assumptions hold and how they connect to standard tools (Holley–Stroock/Bakry–Emery, how to verify mirror LSI), and explained at a high level how the diffusion step may be discretized. The new material is highlighted in blue in the revised pdf. Per reviewer:


- **Reviewer 8nHW (score 8)**
  - Sees MMFLD as a novel and interesting extension of MFLD to constrained domains, with clear potential impact due to the lack of existing methods with convergence guarantees.
  - Asked for more motivating examples unlocked by our analysis, including training weight-constrained two-layer neural networks; we responded by providing corresponding experiments which numerically and visually demonstrate the benefits of MMFLD.
- **Reviewer ZizB (score 6)**
  - Agrees that our paper is "very well-written" and we target a "fairly important and surprisingly open problem", with strong guarantees under standard assumptions.
  - Main concern was that many proof ingredients build on prior work and that the exact diffusion step is idealized. We responded by clarifying that our main contribution is to extend MFLD to boundary-constrained domains for the first time with provable guarantees, and provide a new algorithm which theoretically & practically improves upon existing methods, thus informing practitioners on more stable methods. We also provided a discussion on how the diffusion step may be analyzed.

- **Reviewer qe14 (score 2, indicated willingness to raise to 4)**
  - Agrees the paper is well-written and targets a natural problem by utilizing the full range of available tools.
  - Main concerns were (1) technical novelty, which we addressed above; (2) lack of substantial experiments beyond the toy simplex example. We addressed this in depth by **adding extensive experiments** on (i) PSD matrix optimization in the spectraplex; (ii) classification with norm-constrained ReLU neural networks. The new results numerically and visually demonstrate the **empirical advantages** over projected MFLD (avoiding boundary mass accumulation, improved convergence, reduced overfitting).
  - Also asked for concrete motivation and clearer treatment of LSI and other assumptions; we responded by tying our formulation to key ML applications, expanding the discussion of when LSI and mirror LSI hold, and fixing the technical issues they flagged. Their remaining objection is primarily about the discrete-time bound for barrier mirrors (a known technical issue), not correctness or relevance.

- **Reviewer rL3e (score 4, “happy to raise my score if the authors can address my concerns”)**
  - Requested (i) clearer use of smoothness/LSI assumptions and how constants appear in the theorems, (ii) an experiment on constrained two-layer neural networks, and (iii) tuning step-size and particle number.
  - We addressed these by clarifying the role of the smoothness assumptions, adding the requested constrained NN experiment plus the PSD experiment, and reporting robustness to step size and number of particles. Given that we have satisfied these conditions, we believe this review no longer presents a case against acceptance.


As such, we have thoroughly addressed reviewers' main concerns and believe the current version of the paper is now above the bar for acceptance.

Sincerely, the authors of Paper \#15465

---

### Meta-Review · Area_Chair_9xsR · 2025-12-24

**Summary:**

Reviewers raised a number of concerns:
- The need for the algorithm is not well-motivated, and the experimental validation is weak.
- The novelty is low, as the ingredients are taken from prior works. For example, the propagation of chaos result in Thm. 4.1 requires no new ingredients at all, as it is simply a property of the stationary measures and does not involve the algorithm.
- The theory is not fully satisfactory, as the assumptions seem strong, and the discretization is unrealistic.

I find the authors' assessment that Reviewer rL3e would have raised the score to be unclear (see below). Thus, the submission remains borderline.

From my own reading, I am in agreement that the novelty is low, because the theory does not really attempt to do anything with MMFLD that couldn't be done with MFLD. For example:
- MMFLD assumes uniform mirror LSI. No examples are given in which this is satisfied, except in the case where uniform LSI holds and the mirror map is strongly convex with respect to the Euclidean norm. In that case, one would also have convergence of the original MFLD algorithm?
- Thm. 4.2 assumes that the gradients of the mirror map lie in a bounded set, which contradicts the usual mirror map assumption (that the gradients of the mirror map cover all of $\mathbb R^d$). What are examples of mirror maps and domains that actually satisfy all of the assumptions? Does the theory actually encompass constrained sampling?
Generally, I would have expected the innovation of MMFLD to better handle constrained problems, or problems with geometry different from the usual $\ell_2$ geometry, but without any concrete examples or comparisons with MFLD, this is not well-demonstrated.

The idea of the paper is promising, but in its current state it is rather half-baked. And so, I do not believe it meets the bar.

**Reviewer Concerns:**

Reviewers brought up concerns over motivation, and given that my own thoughts echo these sentiments (lack of concrete examples in which superiority of MMFLD over MFLD is demonstrated), I feel that this is unresolved. Similarly with concerns regarding novelty and theoretical strength.

The authors do provide some additional experimental validation, but it remains weak.

**Reviewer Scores:**

Reviewers 8nHW and ZizB left good scores, and they would have likely remained unchanged.

Reviewer qe14 engaged in discussion with the authors, and indicated a willingness to raise the score to 4.

Reviewer rL3e indicated a willingness to raise the score if the concerns were addressed. The questions regarding which assumptions are needed for which theorems have been addressed. However, the general point about the applicability of the theory due to the strong assumptions has not been resolved. Moreover, the reviewer asked for experiments for two-layer neural networks, and the authors responded with a setup in which the weights are constrained in $\ell_2$ norm. In my view, this does not satisfactorily address the concern: the algorithm is only compared against projected MFLD. To begin with, the constraint on the norms is artificial; standard NN training opts to use weight decay instead of enforcing boundedness of the weights, and weight decay can be implemented within MFLD. So, the experiment does not convincingly demonstrate that MMFLD has any advantages over MFLD in this context. Again, it is not clear to me why the authors focus on an $\ell_2$ problem, when MMFLD is more tailored to handle different geometries, such as $\ell_1$ constraints on the weights which would correspond to the well-known Barron class. With these considerations in mind, I do not believe the reviewer would have changed the score.

---

### Decision · Program_Chairs · 2026-01-26

Reject